TOPICAL REVIEW

# The 70-year search for the voltage-sensing mechanism of ion channels

Luigi Catacuzzeno and Fabio Franciolini

*Department of Chemistry, Biology and Biotechnology, University of Perugia, Perugia, Italy*

Edited by: Ian Forsythe & Thomas DeCoursey

The peer review history is available in the Supporting Information section of this article (https://doi.org/10.1113/JP282780#support-information-section).

**Abstract** This retrospective on the voltage-sensing mechanisms and gating models of ion channels begins in 1952 with the charged gating particles postulated by Hodgkin and Huxley, viewed as charges moving across the membrane and controlling its permeability to $Na^+$ and $K^+$ ions. Hodgkin and Huxley postulated that their movement should generate small and fast capacitive currents, which were recorded 20 years later as gating currents. In the early 1980s, several voltage-dependent channels were cloned and found to share a common architecture: four homologous domains or subunits, each displaying six transmembrane $\alpha$-helical segments, with the fourth segment (S4) displaying four to seven positive charges invariably separated by two non-charged residues. This immediately suggested that this segment was serving as the voltage sensor of the channel (the molecular counterpart of the charged gating particle postulated by Hodgkin and Huxley) and led to the development of the sliding helix model. Twenty years later, the X-ray crystallographic structures of many voltage-dependent channels allowed investigation of their gating by molecular dynamics. Further understanding of how channels gate will benefit greatly from the acquisition of high-resolution structures of each of their relevant

**Luigi Catacuzzeno** received his PhD degree in Cellular and Molecular Biology at the University of Perugia in 2006. He improved his background in theoretical biophysics on membrane ion channels as a PhD fellow at the Department of Physiology and Biophysics at the University of Miami. He is currently an Associate Professor at the Department of Chemistry, Biology and Biotechnology at the University of Perugia. His research interests include cell electrophysiology and ion channel gating and permeation/selectivity mechanisms. His activity includes both experimental and theoretical approaches. **Fabio Franciolini** graduated in Biological Sciences at the University of Perugia. He spent 2 years in Dr Chris Ashley's laboratory at the Department of Physiology, Oxford University, followed by 5 years in Dr Wolfgang Nonner's laboratory at the Department of Physiology and Biophysics, University of Miami. He then returned to the University of Perugia, where he established his own electrophysiology laboratory and continues to research the physiology and biophysics of ion channels.

functional or structural states. This will allow the application of molecular dynamics and other approaches. It will also be key to investigate the energetics of channel gating, permitting an understanding of the physical and molecular determinants of gating. The use of multiscale hierarchical approaches might finally prove to be a rewarding strategy to overcome the limits of the various single approaches to the study of channel gating.

(Received 27 January 2022; accepted after revision 25 April 2022; first published online 4 June 2022)

**Corresponding authors** Luigi Catacuzzeno and Fabio Franciolini: Department of Chemistry, Biology and Biotechnology, University of Perugia, via Pascoli 1, Perugia 06129, Italy.     Emails: luigi.catacuzzeno@unipg.it and fabio.franciolini@unipg.it

**Abstract figure legend** Time line of our understanding of the voltage-sensing mechanisms and gating models of voltage-dependent channels from the charged gating particles postulated by Hodgkin and Huxley in 1952, later visualized as the gating currents, subsequently identified with the fourth segment (S4) as their molecular counterpart within the sliding helix model, then observed in the crystallographic structures of the voltage-dependent channels, and currently investigated by simulations of molecular dynamics.

## Introduction

Our journey begins 70 years ago, in 1952, with the publication of a series of papers in *The Journal of Physiology* by Alan Hodgkin and Andrew Huxley that represented a milestone in our understanding of neuronal excitability (Hodgkin & Huxley, 1952a, 1952b, 1952c, 1952d; Hodgkin et al., 1952). The conceptual and experimental work that underpinned these papers began in the summer of 1947, at the Laboratory of the Marine Biological Association in Plymouth, with Hodgkin and Huxley voltage clamping the giant axons of the squid *Loligo* to record the voltage-dependent $Na^+$ and $K^+$ currents. Kenneth Cole and Howard Curtis (1939) had already described that a change in conductance across the membrane of *Loligo* giant axon occurred during the generation of the axonal action potential, although they did not identify the nature of the underlying conductances. During the following few years at the Physiological Laboratory in Cambridge, they carried out the analysis of these transmembrane currents, their voltage and time dependence, and constructed a mathematical model of the axonal action potential that described accurately its time course, threshold, propagation and refractoriness. This monumental work remains one of the most extraordinary conceptual achievements in biophysics and biology. For their work, Hodgkin and Huxley received the Nobel Prize in Physiology or Medicine in 1963.

How and where these currents crossed the membrane remained unresolved at that time. In their papers, Hodgkin and Huxley referred only to ion currents and conductances, never to ion channels. Indeed, the concept of the ion channel was for the future, and a structural correlate was non-existent. Some would think of holes in the membrane, others of cracks forming upon voltage changes as likely pathways for the ion fluxes. Functional studies in the 1960s, mainly contributed by Bertil Hille and Clay Armstrong, laid the foundation for the notion of $Na^+$ and $K^+$ ions crossing the membrane through aqueous pores provided by specific proteic structures: the ion channels.

## Hodgkin, Huxley and the charged gating particles

In their experiments, Hodgkin and Huxley found that the total membrane current activated by depolarization in the giant axon of the squid could be separated into two components, the $Na^+$ and the $K^+$ currents, $I_{Na}$ and $I_K$. They studied these individually for their dependence on membrane potential and time and, by introducing the parameter conductance ($g_{Na}$ and $g_K$ for $Na^+$ and $K^+$ ions), they linked these currents to their driving force ($V - E_x$) through simple ohmic relationships: $I_K = g_K(V - E_K)$ and $I_{Na} = g_{Na}(V - E_{Na})$. In this way, $g_{Na}$ and $g_K$ could be determined readily because they both coincided with the time course of $I_{Na}$ and $I_K$ in response to a voltage step, under the assumption that during the few milliseconds of the imposed voltage pulses, $E_x$ would not change.

We look first at the time dependence of $g_K$, which is easier to describe. On application of a depolarizing step, we see that it rises slowly, following a sigmoidal time course, until it reaches a steady level. On repolarization, $g_K$ falls exponentially to zero (Fig. 1*A*). A sigmoidal rise of $g_K$ and an exponential fall were suggestive, as Hodgkin and Huxley pointed out, of the $K^+$ conductance being controlled by a number of independent gating particles (Fig. 1*B*), in such a way that all of them had to be in the 'permissive' position for the membrane to pass $K^+$ ions (the sigmoidal rise), but only one of them switching back to the 'non-permissive' position was sufficient to make the membrane impermeant to $K^+$ ions again (the exponential fall). As we said, at the time when Hodgkin and Huxley

proposed their model there was no notion of ion channels as we now know them. Nonetheless, on behalf of novices to the field, from here on we will refer and describe their model in terms of ion channels. We also add that although similar voltage-gating mechanisms are shared by other channels, the scope of this review is restricted to the two channels (conductances) studied by Hodgkin and Huxley.

From fitting $g_K$ at varying voltages, Hodgkin and Huxley found that a good fit was obtained when the number of gating particles, $n$, was made equal to four, giving a probability that all the four gating particles are in the permissive position equal to $n^4$. This notion can be represented analytically by factorizing this parameter in the previous ohmic relationship, to obtain the following equation:

$$I_K = g_{K,max} \times n^4 \times (V - E_K).$$

Given that the opening of $K^+$ channels depended on the membrane potential, the postulated gating particles were assumed to be electrical charges or dipoles that relocate within the membrane in a voltage-dependent manner. Supposing further that each particle moves, in response to voltage, from its non-permissive (resting) to permissive (activated) position and back with first-order kinetics, the distribution of the charged particles as described by the probability $n$ will follow an S-shaped curve ($n^4$), as will the delayed rise of $g_K$ on depolarization, whereas it will fall exponentially to zero, mirroring the decrease of $g_K$ on repolarization.

A similar conceptual approach was used for the $Na^+$ current, with the contingency that the $Na^+$ current turns off spontaneously for the inactivation process. Hodgkin and Huxley accounted for this by introducing the inactivation particle, $h$, that would switch the $Na^+$ conductance off, with maintained depolarizations. With the $Na^+$ current, they obtained a good fit when the number of gating particles ($m$) was made equal to three, giving a probability that all the three particles are in the permissive position equal to $m^3$.

The values of $m$ and $n$ found by Hodgkin and Huxley represent a remarkable prediction that has stood the test of time, to date; $n = 4$ for $g_K$ is correlated with the four subunits forming a typical $K^+$ channel, and the value of

$m = 3$ for $g_{Na}$ is also meaningful, because we now know that the first three domains (DI–DIII) arguably dominate the activation process in $Na^+$ channels, with domain IV apparently modulating inactivation.

With their studies, Hodgkin and Huxley accurately described the ionic currents underlying the action potential and predicted its major properties, including time course, propagation and refractoriness. The following reasons turned out to be crucial for Hodgkin and Huxley to succeed in their endeavour to model the action potential waveform and describe the underlying currents. First was the choice of the squid giant axon that could be voltage clamped effectively using a simple axial wire electrode. Second, they elected to control the membrane potential and measure membrane current on the grounds that membrane conductances to ions were controlled by voltage. Third, the squid giant axon is extremely minimalist in terms of ion currents, displaying essentially only the $Na^+$ and $K^+$ currents that they studied.

Hodgkin and Huxley also proposed key elements of the gating mechanism of the 'future' ion channels when these were only a concept at best: namely, the charged gating particles that are required to move across the membrane electric field to switch on and off $g_{Na}$ and $g_K$ (i.e. to open and close the respective ion channels, in modern terms). This was a signpost to the biophysicists of the time for the direction to go (i.e. towards the gating currents that these charged gating particles would generate with their movement).

## The gating currents

Owing to their small size and fast kinetics and, in addition, being mixed with fast ion currents (such as the $Na^+$ current) and the capacitive currents needed to charge the membrane to new potentials, the first recordings of the postulated gating currents had to wait >20 years after Hodgkin and Huxley had predicted them. They were first recorded for $Ca^{2+}$ channels of skeletal muscles (Schneider & Chandler, 1973) and soon after for $Na^+$ channels of squid giant axons (Armstrong & Bezanilla, 1973; Keynes & Rojas, 1974) (Fig. 2). In the mid-1980s, the cloning

**Figure 1. Hodgkin, Huxley and their charged gating particles**
*A*, $K^+$ conductance time course of a squid giant axon membrane upon depolarization from and repolarization to resting potential. [From Hodgkin & Huxley (1952d).] *B*, sketch illustrating the relocation of the postulated charged gating particles across the membrane upon depolarization. [From Catacuzzeno & Franciolini (2019).]

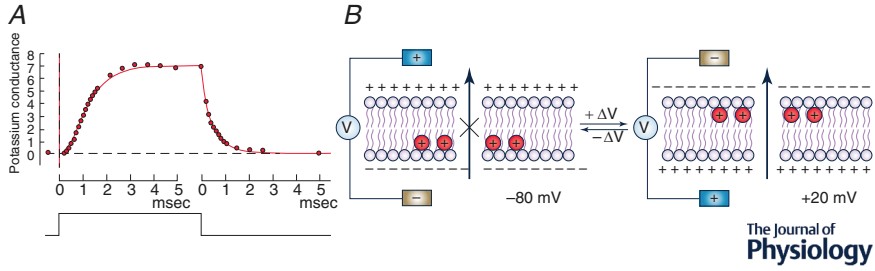

of $Na^+$ and $K^+$ channels and their expression at high density in *Xenopus* oocytes allowed gating currents to be recorded also from $K^+$ channels. This allowed mutations to be inserted into the channel proteins, which started a thorough investigation of the gating current underlying mechanisms. For instance, mutation W434F in the pore region of *Shaker*, which makes the channel inactivate extremely fast, rendered it fully impermeant to $K^+$ ions, while preserving the gating current and its main features virtually unchanged (Perozo et al., 1993; Yang et al., 1997). This mutant channel represented an important tool in the study of the gating currents and became the standard model for investigating the gating currents of $K^+$ channels and the structure–function relationship of its voltage sensor.

**Gating currents neither rise instantaneously nor decay monoexponentially.** As soon as the first gating currents were recorded, their time course demonstrated that the rise was not instantaneous, as expected from Hodgkin and Huxley's two-state model of the charged gating particles, but instead showed a clear rising phase. Likewise, the decay phase could not be described by a single exponential function, as anticipated from the same model, but instead comprised several components (Armstrong & Bezanilla, 1977; White & Bezanilla, 1985). Both the sigmoidal rising phase and the multicomponent decay of the gating currents were thought to depend on more complex kinetics of the gating charge translocation than expected from the two-state model initially proposed. Several states in the activation pathway of the gating particles were later established by modelling experimental gating currents with discrete state rate (Markov) models (Bezanilla et al.,

1994; McCormack et al., 1994; Schoppa et al., 1992; Tytgat & Hess, 1992; Zagotta et al., 1994).

State rate modelling of *Shaker* gating currents revealed several properties of the gating process. First, the four voltage sensors of the channel (the tetrameric architecture of voltage-gated channels had been now established; MacKinnon, 1991) would move for a long portion of their journey mostly independently, as shown in the kinetic scheme of Fig. 2B. It was found that at least three states were needed to reproduce the rising phase, with an initial transition that translocates less charge or has a smaller forward rate constant than the following transitions (Pathak et al., 2005; Smith-Maxwell et al., 1998; Zagotta et al., 1994). On the contrary, there was a high level of cooperativity between the four voltage sensors in the final step that would open the channel. Second, the voltage sensors would move in several steps and with distinctive transition rates to account for the initial rising phase and subsequent multiexponential decay.

Also at variance with the original view was the decay time course of the OFF gating current which, according to Hodgkin and Huxley's model, ought to be three times faster than the ion current decay time (this is because only one particle was required to switch back to the resting position to close a $Na^+$ channel). Experimental data instead showed the gating current and the $Na^+$ current decay with comparable time courses (Bezanilla & Armstrong, 1975). These results indicate that the return of the three independent gating particles to their resting position, associated with the channel closure, follows more complex kinetics.

Taken together, these data show that gating currents are more complex than the two-state charged gating particles initially proposed (when the gating currents had yet to be recorded). The original model clearly required revision in several crucial aspects to account for the new features of the gating currents that were being disclosed (Armstrong & Gilly, 1979)

**Counting the gating charges associated with a single channel opening.** A major parameter that quantifies the voltage sensitivity of an ion channel is the amount of gating charge that needs to move across the membrane to open the channel. This quantity can be estimated from the equilibrium open probability, $p_O$, of a voltage-gated channel, which takes the form of a sigmoidal function of membrane potential (the equation is raised to the fourth power, because it represents four parallel two-state models for the voltage sensor controlling the opening of a channel): $p_O = \{1/(1 + \exp[-z_g \times e(V - V_{\frac{1}{2}})/k_B T])\}^4$. Here, the quantity $-z_g \times e(V - V_{\frac{1}{2}})$ represents the increase in electrical energy associated with the transition of a gating charge from the non-permissive to permissive position, and $z_g$ is the gating charge translocated by a

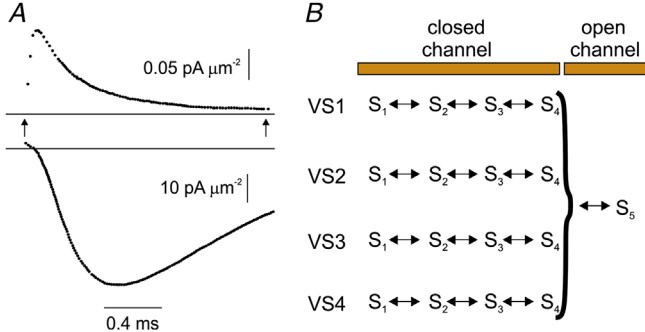

**Figure 2. Gating currents**
*A*, top trace, Armstrog and Bezanilla's first gating current recording from the squid giant axon elicited in response to a 2.5 ms step to 20 mV from −120 mV, in zero external $Na^+$, and with TTX added. *A*, bottom trace, simultaneous recording of $Na^+$ current with the external $Na^+$ concentration reduced to 10 mM. [From Armstrong and Bezanilla (1974)]. *B*, kinetic scheme for the four voltage sensors derived from analysis of the gating current of *Shaker* channels. [From Catacuzzeno et al. (2020).]

single particle that can be measured from the slope of the $p_O$–$V$ curve.

These concepts were originally used by Hodgkin & Huxley (1952d) to gain information on the gating charge associated with the increases in $Na^+$ and $K^+$ conductance, which provided values of 6 and 4.5 elementary charges, $e_0$, that need to move across the whole membrane electric field to open a $Na^+$ and a $K^+$ channel, respectively. They stressed, however, that these ought to be taken as minimum values, because, in principle, that approach is applicable only to a gating charge that translocates between the non-permissive and permissive positions in a single step. Later, several studies demonstrated that the gating kinetics of most voltage-dependent channels, including $Na^+$ and $K^+$ channels, involve more than two states, each associated with the translocation of varying amounts of gating charge. The reasons why two-state models cause the translocated charge to be under-estimated have been provided, for instance, by Bezanilla & Villalba-Galea (2013).

Almers (1978) developed the 'limiting slope method' to assess the effective gating charge, $z_g$, that needs to move to open a voltage-dependent channel also when the movement of the charge occurs in small packages over a sequence of several closed states, thus allowing the applicability of the method to channels with multistate kinetics. The method requires assessment of the $p_O$–$V$ relationship at very negative voltages, where $p_O$ can truly be considered to approach zero, and the $p_O$–$V$ relationship to be linear (the limiting slope), a condition not easily met. Using this method to assess the total charge movement per channel, several groups have reported consistent values of $z_g$ for $Na^+$ and $K^+$ channels ranging between 12 and $16e_0$ (Almers & Armstrong, 1980; Bezanilla & Stefani, 1994; Hirschberg et al., 1995; Zagotta et al., 1994).

Estimates of the gating charge associated with the opening of a single channel can also be obtained from the ratio between the total gating charge, $Q$, moving upon maximal activation (estimated as the time integral of the gating current), and the total number of contributing channels ($N$, estimated with either noise analysis or toxin binding). This method requires independent assessments of $Q$ and $N$ made on the same preparation on which the ratio between the two quantities ($Q/N$) is to be calculated. Several estimates of the gating charge per channel, carried out with this method from $K^+$ channels expressed on *Xenopus* oocytes, were very consistent, falling between $12e_0$ and $14e_0$, and with no apparent bias between the two methods to estimate the number of channels (noise analysis or toxin binding) (Aggarwal & MacKinnon, 1996; Islas & Sigworth, 1999; Noceti et al., 1996; Schoppa et al., 1992; Seoh et al., 1996; Zagotta et al., 1994).

## The primary structure of the $Na^+$ and $K^+$ channels is discovered

In 1984, the voltage-gated $Na^+$ channel from electroplax membranes of *Electrophorus electricus* was cloned and its primary structure determined (Noda et al., 1984). The $Na^+$ channel protein was made of a single polypeptide chain of >1800 residues that folded to form four homologous domains (I–IV), each displaying six transmembrane $\alpha$-helical segments (S1–S6; Fig. 3A). Of particular interest, one segment in each domain (the S4 segment) had four to seven positive charges invariably separated by two non-charged residues. This peculiar and consistent concentration of positively charged residues in the four S4 segments immediately suggested that it might be serving as the voltage sensor in $Na^+$ channels (Greenblatt et al., 1985; Noda et al., 1984). The elucidation of the primary structure of the $Na^+$ channel and the discovery of the highly charged S4 segment led to the development of the sliding helix model to explain the gating mechanism of voltage-gated channels (Box 1).

Shortly afterwards, the *Shaker* $K^+$ channel from *Drosophila* and $Ca^{2+}$ channels from various muscle types were cloned (Kamb et al., 1988; Koch et al., 1990; Mikami et al., 1989; Papazian et al., 1987; Pongs et al., 1988; Tanabe et al., 1987; Tempel et al., 1987), and their overall architecture was found to mirror that of the $Na^+$ channel, with four modules (independent protein subunits for *Shaker* and four domains in a long amino acid stretch for $Ca^{2+}$ channels), each containing six transmembrane $\alpha$-helical segments (S1–S6), of which S4, like S4 of the $Na^+$ channel, contained an excess of positive charges systematically separated by two non-charged residues (see Fig. 3B for *Shaker*), further corroborating the notion that this was the voltage sensor of voltage-gated ion channels.

**Testing the S4 segment as the voltage sensor of voltage-gated $Na^+$ channels.** Conti, Stühmer and co-workers set out to test the primary prediction of the sliding helix model for the voltage-dependent $Na^+$ channel. They reasoned that if the positively charged residues on S4 really are the gating charges of the voltage sensor, their neutralization should reduce the voltage sensitivity of the channel. Using site-directed mutagenesis, they assessed the voltage sensitivity of $Na^+$ channels expressed in *Xenopus* oocytes when one or more positive charges on the S4 segment of domain I were replaced by neutral or negative residues (Stühmer et al., 1989). The reduction by varying amounts of the overall net positive charge induced a corresponding decrease of the apparent gating charge $z_g$ and a rightward shift of the voltage dependence curve. These data supported

## Box 1. The sliding helix model

In 1986, Catterall and, separately, Guy and Seetharamulu converged on the sliding helix (or helical screw) model for the voltage sensor movements during channel gating (Catterall, 1986; Guy & Seetharamulu, 1986). The model depicts the S4 segment, with its positive charges placed along a spiral strip, to be pulled inwards at negative (resting) potential by electrostatic forces, and stabilization of the charged voltage sensor in the hydrophobic environment to be reached by the interaction of the positive charges on S4 with negatively charged residues on nearby helices (Fig. B1). Release of these inwardly directed forces that occurs with depolarization makes the S4 segments move outwards, and the positive charges interact in succession with countercharges on segments S1–S3. Maximal displacement of S4 segments was estimated as 13.5 Å, with a rotation of ∼180°. The interaction of positive and negative charges, also assumed by the model, provided a solution to the thermodynamic dilemma of inserting highly charged structures (the S4 segment) into the extremely hydrophobic environment of the plasma membrane, and the sequential formation of ion pair explained the energetics of the S4 outward movement during activation.

the concept that S4 was the voltage sensor of the Na$^+$ channel, although they did not represent a conclusive demonstration. More compelling experimental data came with the cloning of the *Shaker* K$^+$ channel (Papazian et al., 1987), which was studied more extensively because of its much smaller size (approximately one-quarter that of the Na$^+$ channel), which made it more favourable for genetic manipulations.

The effects of neutralizing the various positive charges on the *Shaker* S4 segment were found to result in a seemingly concurrent decrease of the voltage sensitivity of the K$^+$ current, as first reported by Logothetis et al. (1993). Some inconsistencies emerged, however, because it was found that mutations of positive charges at different positions could have very diverse effects, indicating that the voltage dependence of the channel could not be fully explained by electrostatic considerations only. Two groups addressed this issue directly by assessing the gating charge per channel after charge neutralization on the

*Shaker* S4 segment (Aggarwal & MacKinnon, 1996; Seoh et al., 1996). They both found a significantly lower gating charge translocation upon channel activation when they neutralized the first four S4 residues, R1–R4 (in fact, Seoh et al. (1996) were not able to assess the contribution of R1 to channel gating), in comparison to K5 or K7, which showed only a little difference from the wild-type. [Neutralization of residue R6 (R377) resulted in no viable channels in the plasma membrane. However, a later study with histidine scanning mutagenesis showed that R377 does not participate in gating; Starace & Bezanilla, 2001.] Overall, the data from these two studies indicate that the first four basic residues (R1–R4) are those principally involved in generating the gating current, and thus important in gating (Aggarwal & MacKinnon, 1996; Bezanilla, 2002; Seoh et al., 1996).

A second expectation of the sliding helix model was that the gating charges on S4, while moving upwards through discrete states upon channel activation, would

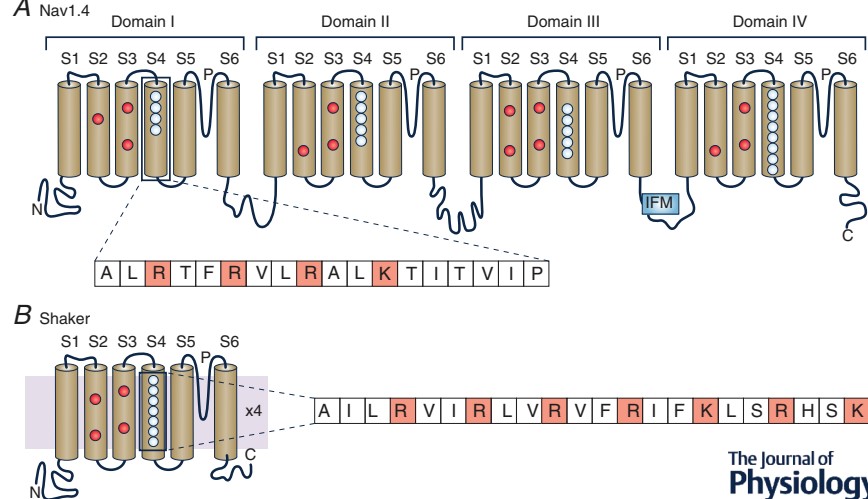

**Figure 3. Primary structures of the voltage-gated human Nav1.4 and *Shaker* K$^+$ channels**
Typical six-transmembrane α-helical segments (S1–S6) architecture of each domain (of Nav1.4; *A*) or subunit (of *Shaker*; *B*) of voltage-gated channels, showing the S4 segment to contain an excess of positive charges invariably separated by two non-charged residues (see sequences of S4 segments in insets). Basic residues on the S4 segment are shown in blue, and acidic residues on S2 and S3 in red. [Modified from Bezanilla (2008).]

generate miniature currents. The size of these 'shot' currents, as they came to be known, was below the resolution of electrophysiological instrumentation. They would, however, create fluctuations on the gating current recorded from a population of channels, and the analysis of these fluctuations could provide information on their features. Conti and Stühmer expressed rat brain Na$^+$ channels in *Xenopus* oocytes at a high density and analysed the fluctuations (noise) of the gating currents. By taking this approach, they found that: (i) the auto-correlation of the fluctuations was consistent with a shot-like type of movement of the charged gating particles (thus excluding the possibility that the charged particles move in a diffusional regime); and (ii) the movement of the gating charge when a single Na$^+$ channel is activated displays a major step carrying an equivalent of $\sim$2.3e$_0$. A few years later, Bezanilla's group, taking a similar approach, found similar values for the *Shaker* K$^+$ channels (2.4e$_0$; Sigg et al., 1994). These authors suggested that this large quantal charge was possibly associated with a late step in the S4 translocation upon activation.

On re-examining Conti & Stühmer (1989) data on Nav channels in the attempt to generalize their conclusions to other voltage-gated channels, Crouzy & Sigworth (1993) suggested that the 2.3e$_0$ obtained by Conti and Stühmer, and likewise the 2.4e$_0$ later obtained by Sigg et al. (1994), might not reflect the true size of the charge crossing the pore. Owing to the limited filter bandwidth used in Conti and Stühmer's experiments (8 kHz), a rapid passage of consecutive unitary (gating) charges through the pore would not appear as single passages, but would become indistinguishable from a single large charge movement; in other words, an experimental artefact attributable to limited filter bandwidth.

**The sliding helix model predicts that S4 moves outwards upon activation.** Voltage-dependent movement of S4 was tested initially on Na$^+$ channels by using the substituted cysteine accessibility method, which involves replacing specific residues on S4 with cysteine, and probing whether cysteine-binding compounds, such as methanethiosulphonate (MTS), can reach these cysteines from either side of the membrane, as a means of defining the position of S4. The binding of MTS to substituted cysteines could be visualized as modifications of voltage-dependent gating. In the first group of experiments, cysteine was substituted for the first (most outward) arginine residue, R1 (R1448), in S4 of domain IV of skeletal muscle Na$^+$ channels (Yang & Horn, 1995). External application of MTS did not affect the Na$^+$ current when the membrane was held at negative (resting) potentials. In contrast, it drastically modified the Na$^+$ current kinetics (the inactivation rate) when the membrane was depolarized, as would be expected for a voltage sensor moving outwards with depolarization (Yang & Horn, 1995). In further studies, cysteine replaced arginine at positions R2 and R3 of the same Na$^+$ channel segment, S4. Cysteines at these positions could be accessed by MTS reagents applied intracellularly when the cell was hyperpolarized, whereas they were reached from the outside after depolarization (Yang et al., 1996).

A similar approach was used in Isacoff's laboratory to probe the movement of S4 of *Shaker* channels upon activation (Baker et al., 1998; Larsson et al., 1996). They replaced several residues, including three gating charges (R362, R365 and R368), over a stretch of 18 amino acids (from A359 to S376) of the *Shaker* S4 region and tested their intracellular and extracellular thiol reagent accessibility as a function of voltage. Large distal portions of S4, with the exclusion of only a short central part, were found to be accessible to the thiol reagent, indicating that S4 can slide into the internal and external vestibules in a voltage-dependent manner. Taken together, these results indicate that at hyperpolarized (resting) potentials S4 protrudes into the intracellular vestibule with all but the first gating charge (R1), whereas it moves outwards, upon depolarization, to expose the three outermost gating charges, R1–R3, to the external vestibule. These studies

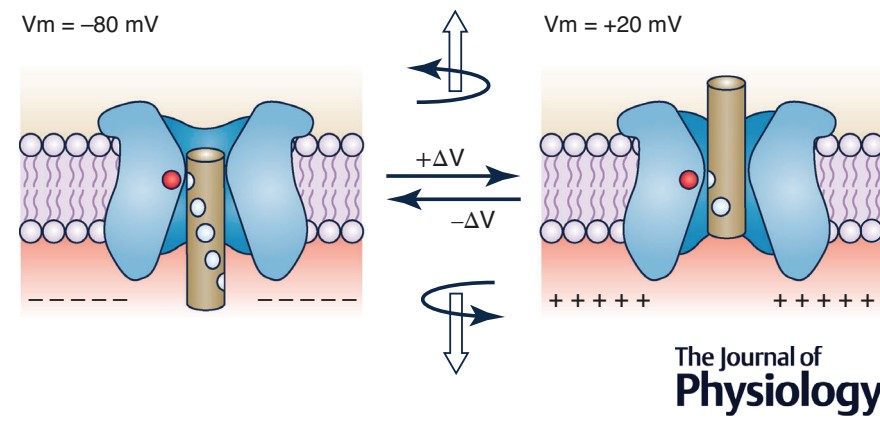

**Figure B1. The sliding helix model**
Sketch of a modernized version of the sliding helix model suggesting that the positive charges, arranged in a spiral shape, pull the S4 segment inwards in the resting state (at negative potential inside). After activation, the inwardly directed forces on S4 are released and the segment is pushed outwards, allowing the positive charges to pair in succession with the negatively charged residues on the vestibule walls (shown as a red dot). [From Catacuzzeno et al. (2020).]

gave indirect estimates of the S4 translocation upon activation in the range 8–15 Å.

To assess the S4 movement during gating rigorously, Bezanilla and co-workers used the lanthanide-based resonance energy transfer (LRET) technique, which can measure accurately the distance between two light-sensitive molecules even when they move relative to each other by only a few ångströms (Cha et al., 1999). They initially attached the chromophores at similar positions on the S4 of different subunits of *Shaker*. With this arrangement, they recorded LRET signals during voltage gating consistent with an S4 rotation of 180° (Cha et al., 1999). The same amount of S4 twisting during *Shaker* channel activation was found by Isacoff's laboratory from the analysis of the fluorescence resonance energy transfer (FRET) experiment carried out to measure the intersubunit distances between different S4 segments of the channel, in both resting and activated states (Glauner et al., 1999). Based on their spectroscopic data, they poposed that the 180° rotation of S4 could account for much of the whole gating charge displacement across the membrane electric field, with only a minimal axial translocation.

To provide a more reliable estimate of the effective translocation of S4 with respect to the surface normal to the membrane, years later Bezanilla's laboratory made FRET measurements between one chromophore attached to a toxin bound to the pore mouth of the channel and the other chromophore attached to the upper section of S4 (Chanda et al., 2005). These experiments showed a translocation of S4 of <2 Å, clearly in contrast to both the sliding helix model and many experimental data. These results called for some new mechanism of gating that could explain how such a small S4 translation could account for the $12-14e_0$ per channel translocated with full channel activation (Box 2). Additional FRET experiments, probing a higher number of residues, indicated that the translocation of the voltage sensor was, in fact, significantly higher and closer to 10 Å (Posson & Selvin, 2008), a result more consistent with a large array of data and the sliding helix model.

### X-Ray crystallographic pictures of K⁺ and Na⁺ channels

By the turn of this century, the basic principles of voltage-dependent gating had been consolidated: the S4 segment had been demonstrated to be the voltage sensor; the positively charged amino acids crucial for channel gating (R1–R4) had been identified; the number of gating charges ($12-14e_0$) that move across the membrane electric field (by $10-15$ Å) during channel gating had been established; and the general movement of the voltage sensor through the gating pore and the zipper-like

### Box 2. The transporter model

To accommodate their FRET results, Bezanilla and co-workers proposed the transporter (rotational) model, which pictures the aqueous crevices from both sides of the membrane penetrating deep into the voltage-sensing domain (VSD), in such a way as to shape the electric field around the S4 segment more parallel to the plane of the membrane (Fig. B2). According to the model, the first four gating charges on S4 (R1–R4; those important for channel gating) are exposed to the narrow aqueous intracellular crevice in hyperpolarized conditions, but to the extracellular crevice upon depolarization, as result of a 180° rotation of S4, with a very minor translation relative to the plane of the membrane. Assuming a focused electric field across the S4 segment, the model was able to account for the $12-14e_0$ moved across the electric field upon VSD activation without any significant translational movement of the helix (Bezanilla, 2002; Cha & Bezanilla, 1998; Cha et al., 1999).

interaction with voltage-sensing domain (VSD) walls had been determined. The next step was to extend these findings with high-resolution structural evidence, which in the early 2000s began to appear as X-ray crystallographic three-dimensional molecular structures of several voltage-gated channels. The research activity on voltage sensing in ion channels in this period was rewarded by unprecedented advances.

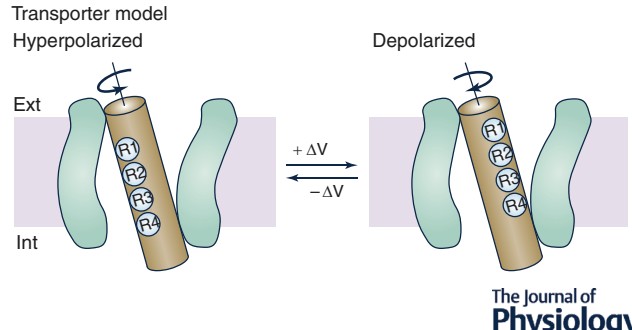

**Figure B2. The transporter model**
The first four charges of the S4 segment (R1–R4) are exposed in hyperpolarized conditions (left) towards the narrow aqueous intracellular crevice, but in depolarized conditions (right) towards the extracellular crevice. According to the model, the depolarization translocates the four gating charges from the internal crevice to the external crevice, with a ~180° rotation of S4, without any significant translation normal to the plane of the membrane. The model requires a highly focused electric field between the internal and external crevices.

## Box 3. The paddle model

In the KvAP, the S4 segment was found tightly bound to the S3b (the second half of segment S3, which is bent in the middle) to form the S3b–S4 helical hairpin, and this paired structure was found dislocated deep inwards near the protein–lipid interface and nearly parallel to the surface of the lipid bilayer (Fig. B3). These observations, combined with structural data of the isolated voltage-sensing module showing interactions of the innermost gating charges with negative groups in the lower portion of S2 (Jiang, Lee, et al., 2003; Jiang, Ruta et al., 2003), suggested a new gating mechanism (the paddle model of activation), in which the S4 segment, during activation, moves considerably through the phospholipid matrix to translocate the estimated $12-14e_0$ across the membrane electric field (Fig. B3).

**Crystallographic structure of the KvAP channel.** The first X-ray crystallographic structure of a voltage-gated ion channel was reported by MacKinnon's laboratory for the bacterial $K^+$ channel KvAP, from the thermophilic *Aeropyrum pernix* (Jiang, Ruta et al., 2003; Ruta et al., 2003). It confirmed the tetrameric architecture of the voltage-gated $K^+$ channels derived from functional studies, with the S5 and S6 segments of the four subunits forming the permeation pore, and the S1–S4 segments making up the VSD. The four VSDs, which displayed both sequences and properties similar to their eukaryotic counterparts, appeared located at the periphery, loosely joined to the pore domain by S4–S5 linkers. This structure was unexpected in several respects, however, and inspired a voltage-gating model distinct from both the sliding helix and transporter models (Box 3).

Later structures of $K^+$ channels showed an architecture different from the KvAP, which made scientists think that the KvAP channel had been significantly altered by the crystallization procedure, such that it no longer represented its native state (Lee et al., 2005). The most likely reason underlying the distorted structure of KvAP was it being crystallized in the absence of lipids and with an antibody bound to it (Lee et al., 2005; Long et al., 2005a, 2005b, 2007).

**The crystal structures of Kv1.2 and Kv1.2/2.1 chimera.** The crystal structure of KvAP was soon followed by the elucidation of the structure of the voltage-gated $K^+$ channels Kv1.2 (Long et al., 2005a, 2005b) and the Kv1.2/2.1 chimera, composed of the Kv1.2 channel and the voltage sensor from the Kv2.1 channel (Long et al., 2007). These structures presented the same architecture as KvAP, highlighting the key elements for channel gating

and reinforcing the findings of previous studies. Namely, the S1–S3 segments form an hourglass-shaped structure with a central gating pore, a short and narrow hydrophobic constriction (or hydrophobic seal; HCS) essentially impermeant to water and ions that separates the internal and external aqueous vestibules and serves to focus virtually the whole transmembrane electric field to a range of only a few ångströms (Chanda et al., 2005; Starace & Bezanilla, 2004; Tombola et al., 2005; Yang et al., 1996).

In the Kv1.2/2.1 chimera, whose VSD structure is shown in Fig. 4*A*, the high-resistance hydrophobic seal has a phenylalanine at 233 (F233) as a key residue (F224 in Kv1.2). On both sides of the gating pore, the Kv1.2/2.1 chimera VSD displays two negative charge clusters, one on the extracellular side (Extracellular Negative Cluster - ENC formed by E183 on S1 and E226 on S3) and one on the intracellular side (Intracellular Negative Cluster - INC formed by D259 on S3 and E236 on S2; Fig. 4*A*). These charge clusters were much expected, as already postulated in the sliding helix model to serve as transitory ion partners for the movement of the charged voltage sensor, S4. The crystal structure also shows that R4 interacts with the ENC (namely with E226), while the negative residues (D259 and E236) of the INC trap the gating charge, K5. This arrangement indicates that crystallization captured the VSD of Kv1.2/2.1 chimera in the activated state, i.e. strongly pushed outwards, with the four gating charges (R1–R4) on the external side of the hydrophobic seal, as expected given that the crystallization procedure cancels the potential across the membrane.

**The gating charge transfer centre model.** Reflecting on the crystal structure of the Kv1.2/2.1 chimera, with the first four arginines of S4 (R1–R4) on the external side of the hydrophobic seal and the next gating charge, lysine (K5), trapped in a putative binding site made by the evolutionarily conserved phenylalanine F233 and the negative charges of glutamate and aspartate on S2 and S3 (E236, D259; Fig. 4*B*), MacKinnon and co-workers thought that this peculiar and conserved grouping of

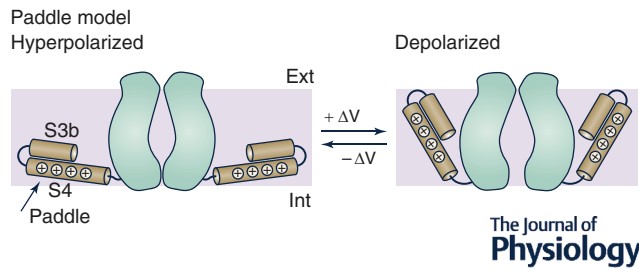

**Figure B3. The paddle model**
Schematic side views of the KvAP channel showing the depolarization-driven outward translocation of S4–S3b paired segments (the paddle) that results in the opening of the channel.

## Box 4. The gating charge transfer centre model

The model maintains that before crossing the gating pore, the gating charges on the S4 segment bind transiently to the gating charge transfer centre (GCTC), made of the aromatic phenylalanine (F290) and the two negative residues E293 and D316 (Fig. B4; Tao et al., 2010). By solvating and stabilizing the gating charges in succession, the GCTC would facilitate the movement of S4 during gating; that is, it would stabilize the S4 gating charges transiently before traversing the gating pore. According to this view, the first gating charge, arginine R1, ought to be found in the GCTC in the fully resting state (at very negative voltage), whereas when fully activated (at very positive voltage), lysine K5 should sit in the GCTC.

charged amino acids would have to serve a major role in gating. To address this point, they mutated the phenylalanine of the *Shaker* VSD with several substitutes and measured the resulting K⁺ currents from *Xenopus* oocytes, the expression system used for the mutated channels (Tao et al., 2010). As result of their investigation, they proposed the gating charge transfer centre (GCTC) model (Box 4).

The role of GCTC in channel gating was challenged by Ahern's laboratory based on the observation that a neutral synthetic substitute at either E293 or D316 of *Shaker* did not appreciably modify the conductance–voltage curve (Pless et al., 2011). The two residues E293 or D316 were, in addition, suggested to be located in the water-filled

intracellular vestibule, thus outside the transmembrane electric field, greatly weakening their proposed role. On this basis, they suggested that some role in assisting channel gating could be played only by the highly conserved phenylalanine, by establishing transient gating charge–π interactions during S4 movement (Pless et al., 2011).

The GCTC notion was also questioned on the grounds that it was based essentially on the effects of F290 mutants on the K⁺ currents, although ion currents are known not exactly to mirror what is going on at the voltage sensors. Lacroix and Bezanilla verified the effects of F290 mutants on the gating currents of *Shaker*; that is, the direct object of the action of the mutants. Of the 13 mutations tested, only one (F290W) exhibited a meaningful effect on the gating charges (Lacroix & Bezanilla, 2011).

These experiments rolled back the enthusiasm for the GCTC view. However, when the first crystal structure of a bacterial Na⁺ channel was disclosed, a structure absolutely similar to the GCTC of *Shaker* was also found in this channel (Payandeh et al., 2011). Specific tests to address its function suggested, however, that the negative residues in the Na⁺ channel GCTC seemed important for VSD structuring and channel trafficking to the membrane, leaving its role open to question (see next paragraph entitled "The first crystallographic structure of a voltage-gated Na⁺ channel is revealed").

**The first crystallographic structure of a voltage-gated Na⁺ channel is revealed.** In 2011, Catterall's laboratory reported the first high-resolution crystallographic structure of a voltage-gated Na⁺ channel (Payandeh et al., 2011). The channel (the bacterial NavAb from

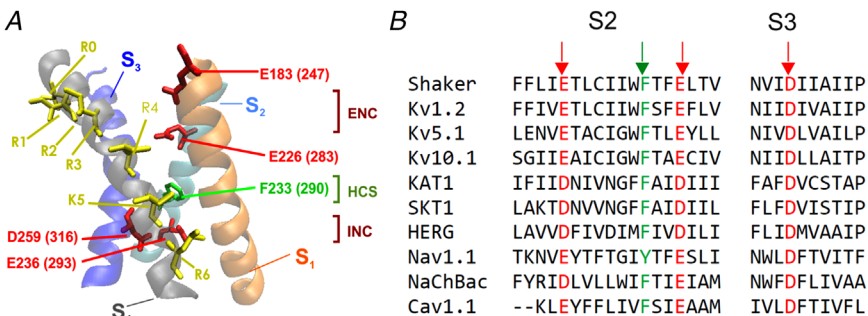

**Figure 4. Ribbon representation of the four α-helical structures of one voltage-sensing domain of Kv1.2/2.1 chimera in the activated state**

*A*, stick representation of the positive gating charges on the S4 segment (R0–R4 and K5, in yellow), the negative residues E183 on S1 and E226 on S3 (red), forming the Extracellular Negative Cluster (ENC), and the Intracellular Negative Cluster (INC), made by the residues D259 and E236. The central part of the voltage-sensing domain, forming the gating pore, or Hydrophobic Constriction Site (HCS), is essentially contributed by a phenylalanine residue (F233, green). Given that this structure is common to several K⁺ channels, in parenthesis we give the corresponding residue numbers found in the *Shaker* channel, in which much investigation has been carried out. [From Catacuzzeno et al. (2020).] *B*, sequence alignment of S2 and S3 from various channels. Highlighted are the conserved residues investigated here.

*Arcobacter butzleri*) was made of four identical subunits, similar to those of voltage-gated K$^+$ channels, and each subunit mirrored a single domain of mammalian Na$^+$ channels, with the six transmembrane segments making the VSD (S1–S4) and the pore domain (S5 and S6). The S1–S3 segments of the VSD were found to form an hourglass-shaped structure, along the lines of the VSD of K$^+$ channels, with a short and narrow hydrophobic constriction (hydrophobic seal or plug) in its central part, essentially impermeant to water and ions, that separates the intracellular and the extracellular aqueous vestibules and focuses virtually the whole transmembrane electric field (Yang et al., 1996). This hydrophobic seal again includes a phenylalanine as its most characteristic residue, showing its high evolutionary conservation. On both sides of the hydrophobic seal, the NavAb VSD displays two negative charge clusters that earlier disulphide cross-linking studies using paired substituted-cysteine residues had already shown on a Na$^+$ channel homologue of NavAb, the NaChBac from *Bacillus halodurans*.

The highly conserved internal cluster, made of the aromatic phenylalanine F56 and the two close negatively charged D80 and E59, has been suggested to be implicated, as in the *Shaker* channel, in the stabilization of the S4 charges, while the S4 segment steps up through the gating pore during activation. The proposed function of these structures of transiently binding the gating charges on S4 before they cross the gating pore during activation has been questioned, however, on the grounds that in the Nav1.4 channel these residues have been reported to serve, as indicated above, for its proper folding and membrane trafficking. An alternative function proposed for these structures is to shape the potential profile across the membrane and, as result, establish the kinetics of

the S4 motion. In particular, the wider hydrophobic seal of the VSD of domain IV of the Nav1.4 channel compared with the other three VSDs, as reported by Gosselin-Badaroudine et al. (2012), has been taken to explain its slower activation (Chanda & Bezanilla, 2002).

In concluding this section, we are mindful that structural data taken by themselves can often be misleading when translated directly into functional terms. Channel structures obtained by crystallization can provide only a poor reflection of the real structure of the native channels in the cell membrane (owing to preparatory procedures for protein crystallization that rarely, if ever, preserve the natural biological environment). Even if the native environment of channels is conserved, there are thermodynamic considerations when it comes to understanding what the structural model in question represents. For instance, a recent structure of the human Kv7.2 associated with calmodulin shows activated voltage sensors with a likely closed pore for a channel that does not inactivate (Li et al., 2021). Conversely, the hERG channel structure, found with an activated voltage sensor, displays an apparently conducting selectivity filter, although this channel undergoes a rapid inactivation at depolarized potentials (Wang & MacKinnon, 2017). Considering another example, the VSD of channels undergoes voltage-independent transitions following voltage-dependent ones (Villalba-Galea & Chiem, 2020). The existence of such transitions, well documented yet largely ignored for several decades, implies that the 'open' state is likely to be meta-stable and hardly attainable with the current approaches taken in structural biology. These examples lead to the conclusion that channels adopt conformations beyond the simple opening and closing of their pores, and extreme care needs be exercised when interpreting or using structural data. Some help for

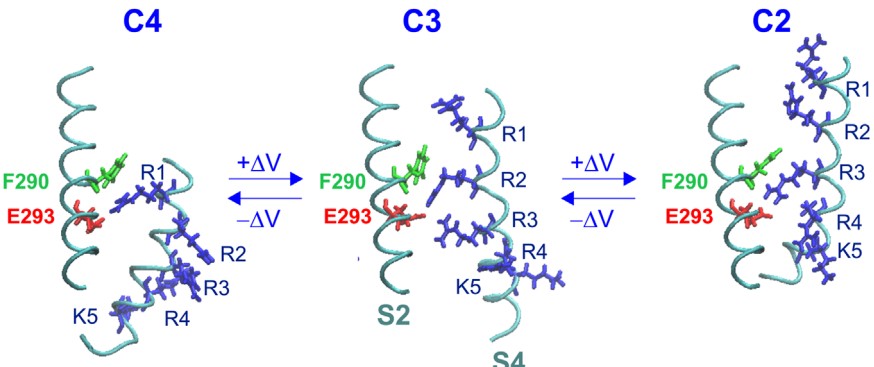

**Figure B4. The gating charge transfer centre model**
Scheme showing the gating charge transfer centre (GCTC) hypothesis, with each state characterized by a different gating charge present in the GCTC formed by the aromatic phenylalanine at 290 and the negative residues E293 and D316. For clarity, the negative residue D316 on S3 has been omitted. The three states shown (C4, C3 and C2) are from the study by Henrion et al. (2012), with C4 representing the 'deep closed state' at most negative potentials, with the first gating charge (R1) in the GCTC.

interpreting the experimentally available structures might certainly come from atomistic simulation techniques that we are about to introduce.

## *Ab initio* and molecular dynamics simulations to picture the closed state and the voltage sensor movement

Given that virtually all the crystal structures reported so far portray voltage-gated channels in the open/inactivated state, structure-based atomic simulations have been used recently to understand the structure of the resting VSD, including *ab initio* modelling and molecular dynamics (MD) simulations.

**The resting state structure of Kv1.2 VSD from *ab initio* modelling and MD simulations.** The first model of a voltage-gated channel in the resting state derived from structural data was obtained by Catterall's laboratory using *ab initio* Rosetta modelling on the Kv1.2 channel structure (Yarov-Yarovoy et al., 2006). The model was based on structural restraints and established experimental data, such as the distance between the first gating charge (R1) and the negative residues E226 on S2 and D259 on S3, or the exposure of the gating charges R3 and R4 to the intracellular water-accessible vestibule. The outward movement of the S4 segment that was produced by VSD activation from the closed state model to the activated state of the crystallographic structure was incredibly small, only ∼3 Å (and a clockwise rotation of ∼180° on its axis; Yarov-Yarovoy et al., 2006). A similar *ab initio* modelling procedure, in conjunction with MD simulations, was used by Isacoff's laboratory to define the resting state of Kv1.2 VSD (Pathak et al., 2007). In addition to the standard experimental constraints, they took into account data from fluorescence scans on the local motion of external portions of the *Shaker* channel protein associated with specific activation steps. Although the overall conclusions on the voltage sensor activation mechanism were similar to those proposed by Catterall's laboratory, the results obtained by Isacoff and co-workers differed in predicting a significantly larger S4 translocation (6−8 Å).

Starting from the closed state model of Kv1.2 developed in Isacoff's laboratory, Khalili-Araghi et al. (2010) made MD simulations in an explicit membrane and surrounding solvent, including polar lipid headgroups, to refine the resting state model. In all simulations, they found the formation of salt bridges between the gating charges on S4 and the highly conserved negative charges in S2 and S3 (interactions with polar lipids at the membrane–solution interface were also found). Their simulations also indicated a more inward position for S4 in the resting state than previously suggested by

Isacoff and co-workers (Pathak et al., 2007). Given that the focused transmembrane electric field fell within a distance not larger than ∼10 Å, owing to the water-filled vestibules on both sides of the water-impermeant hydrophobic plug, they estimated a movement of S4 of ∼7 Å as sufficient to account for a charge of $12-14e_0$ transported across the membrane upon full VSD activation.

Khalili-Araghi et al. (2010) also found that a portion of S4 $\alpha$-helix containing ∼10 residues, located across the catalytic centre, spontaneously transformed to a 3.10-helix while reaching the resting conformation. The 3.10-helix conformation would bring the gating arginine residues (R1–R4) on S4 to align vertically on one face of the helix, favouring salt bridge formation with acidic residues in S2 and S3 and energetically stabilizing the resting state. The presence of a 3.10-helix stretch in the lower portion of S4 (∼11 residues) had been observed already in the open-state X-ray structures of Kv1.2 channels (Long et al., 2007). Catterall's group also observed the S4 segments to adopt a 3.10-helix conformation in the bacterial Na$^+$ channel NaChBac and suggested that charge pairing between the gating charges and the acidic residues forming the external and internal negative clusters that occur while S4 passes through the catalytic centre would be facilitated by a transient shift of a stretch of S4 from $\alpha$-helix into a 3.10-helix (DeCaen et al., 2009). A significant portion of the S4 segment was also found in the 3.10-helix conformation by Lindahl's group when they modelled with all-atom molecular dynamics the transition from the open X-ray structure of the Kv1.2/2.1 chimera to the resting state, under the influence of a negative membrane potential (Bjelkmar et al., 2009). Lindahl and co-workers also showed that the 3.10-helix transition that occurred between Q1 and R3 reduced the free energy associated with the initial steps of S4 movements towards the resting state (Schwaiger et al., 2011).

Using MD simulations on the Kv1.2 channel, Delemotte et al. (2011) found a sequential translocation of the voltage sensor gating charges through the GCTC, characterized by its highly conserved phenylalanine F233, to reach the resting state of the channel that was characterized by all the gating charges having slipped below F233. Similar results were reported by Elinder and co-workers using the Cd$^{2+}$ bridges strategy on the *Shaker* channel to estimate the position of S4 and its interactions with the other segments of the VSD in several states, including the closed state(s) (Henrion et al., 2012). They found that in very hyper-polarizing conditions, all the gating charges, including R1, became stabilized below F290. At a slightly milder hyper-polarization, R1 was instead found above F290, interacting through the salt bridge with E283.

These resting state models were found to converge strongly towards a structure wherein all the gating charges on S4 were pushed inwards below the hydrophobic plug,

except for arginine R1, which remained above F290, stably interacting with E1. We should recall that MacKinnon's laboratory had instead placed R1 below F290 in the resting state of *Shaker*, stabilized into the catalytic centre (GCTC) by the acidic residues E293 (E2) in S2 and D316 (D3) in S3 (Tao et al., 2010; Fig. 5*A*). Their view found experimental support from the $Zn^{2+}$ bridging experiments in the *Shaker*, where the double mutant I287H on S2 and R1H on S4 would allow the formation of $Zn^{2+}$ metal bridges in the resting state, suggesting that R1H had shifted further down, past F290, and bound to the GCTC (Lin et al., 2011).

This discrepancy among different studies with regard to the position of R1 at rest might be attrbutable to the presence of more than one 'resting' state, each occupied in relationship to how much negative voltage is applied, and possibly different in the various studies. According to this view, the consensus model of Vargas et al. (2011) might represent the resting state more populated at intermediate hyperpolarizations (the 'penultimate resting state', as it was termed by Lin et al., 2011), whereas the resting state reported by Tao et al. (2010) and Delemotte et al. (2011), with R1 in the GCTC, can be reached only with strong hyperpolarizations (Fig. 5*B*). An additional consideration that could help to reconcile these differing results is that most of the studies placing R1 above F290 were carried out upon mutating R1 into a neutral residue. It can be argued that after R1 neutralization, the S4 segment can hardly be able to reach its most inward position because it is only on the charged R1 where the transmembrane voltage acts to pull S4 fully inwards.

**The first resting structures of the VSD of the Na$^+$ channel are provided by cryo-electron microscopy.** In 2019, Catterall's laboratory reported the high-resolution cryo-electron microscopy (cryo-EM) and X-ray structure of the resting state of the bacterial Na$^+$ channel NavAb,

which they stabilized in this state by introducing mutations that were previously shown to shift the activation $V_{\frac{1}{2}}$ of the Na$^+$ channel to +60 mV, with the result that at 0 mV, the condition experienced during the cryo-EM procedure, the channel was in the resting state (Wisedchaisri et al., 2019). Moreover, they introduced disulphide crosslinks to lock the channel structure into the desired [resting (mutant channels) or activated (wild-type channels)] conformation (Gamal El-Din et al., 2013; Lopez et al., 1991). The stabilization of these states in these mutants was verified in functional studies to ensure that the structural data have clear functional correlates.

Analysis of these constructs provided crucial insights into the structural rearrangements associated with the gating transitions of the VSD of the channel from the resting to the activated state, and back. These included an inward displacement of ~11.5 Å for the S4 segment and a significant rotation, with the gating charges interacting with different ion pair partners. In the resting state, the first gating charge (R1) was found above the gating pore. With the resting and activated states of NavAb available, Catterall and co-workers could also model the putative transitions during channel activation. It was suggested that the outward movement of S4 occurring during activation translocates three gating charges (R2, R3 and R4) through the hydrophobic plug/transmembrane electric field. These conformation transitions were compatible with the classical sliding helix model, in which the S4 segment moves while rotating along its axis. The gating charges, not directly exposed to the lipid hydrocarbon, interacted by forming sequential salt bridges with the acidic intracellular carboxyl-terminal domains, as originally proposed by Clay Armstrong (1981). Moreover, the transmembrane potential driving the translocation of gating charges through the gating pore is concentrated over a narrow

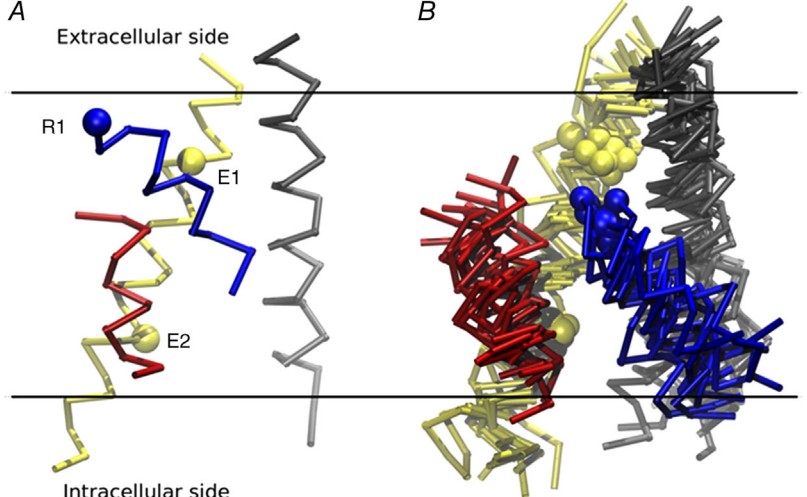

**Figure 5. The consensus model of the resting state of the voltage-sensing domain of Kv1.2 channels**

*A*, voltage-sensing domain of Kv1.2 in the active state, shown for reference. The spheres represent the Cα atoms of E1 (E226) and E2 (E236) on S2, and R1 (R294) on S4. *B*, superimposed resting state voltage-sensing domain models from Delemotte et al. (2011); Henrion et al. (2012); Jensen et al. (2012); Pathak et al. (2007); and Vargas et al. (2011). Colour code of the four helices: S1, grey; S2, yellow; S3, red; and S4, blue. [From Vargas et al. (2012).]

region of $\sim$10 Å, which includes the evolutionarily conserved phenylalanine residue (Wisedchaisri et al., 2019).

**The five-state gating model.** Using a combination of MD simulations and biased MD to visualize the conformation changes of the Kv1.2 VSD during channel deactivation (i.e. at hyperpolarized potentials), Delemotte et al. (2011) found five available states that the channel could occupy during the deactivation process (Box 5). In addition to the open (starting) state ($\alpha$) and the resting (final) state ($\varepsilon$), the channels could dwell in three intermediate states ($\beta$, $\gamma$ and $\delta$). While moving, the S4 segment would sequentially establish and break ion pairs with nearby negatively charged residues and lipid head groups, in a zipper-like fashion, as previously suggested. The positions of the gating charges on S4 in the resting ($\varepsilon$) state were congruent with those found by Pathak et al. (2007), using *ab initio* modelling, and by Khalili-Araghi et al. (2010), obtained with MD simulations. They also assessed the gating charge translocated during the entire transition process, from the open to the closed state, and found it to amount to $\sim$12.0$e_0$, in good agreement with major biophysical studies (12−14$e_0$; Aggarwal & MacKinnon, 1996; Schoppa et al., 1992; Seoh et al., 1996). The simulations also showed that in one of the four VSDs, the passage of the gating charges through the catalytic centre involved the switching of a short stretch encompassing the charged residue that is crossing the gating pore into a 3.10-helix conformation, in a manner similar to the proposition advanced by Catterall's group (DeCaen et al., 2009). For the other VSDs, the passage of charges through the catalytic centre occurred without any structural change of the $\alpha$-helix conformation of the S4 segment (Delemotte et al., 2011).

Elinder's laboratory made a significant contribution to the ongoing debate by presenting a whole voltage sensor gating cycle composed of one open and four closed states (Henrion et al., 2012; Box 5). They studied 20 specific interactions between S4 and the other segments of the VSD, in the framework of the five states that resulted from metal ($Cd^{2+}$) ion-bridge studies with cysteines. Models for each state were generated by Rosetta modelling (with no assumptions on the spatial position of the VSD helices from other studies), and finally, refined by repeated MD simulations (Henrion et al., 2012). According to this study, the S4 segment moves by no less than 12 Å from the active state (O) to the C3 state (the equivalent of the $\delta$ state of Delemotte et al., 2011), in this motion shifting three charges across the full membrane voltage drop. A deeper resting (closed) state (C4) of the VSD (the $\varepsilon$ state of Delemotte et al., 2011) could be reached with very large hyperpolarizations (in which case, the S4 would move by $\sim$17 Å). As already reported, Henrion et al.

(2012) also found the S4 $\alpha$-helix transiently switching into a 3.10-helix, over a segment of $\sim$10 amino acids that maintains its location across the hydrophobic region, i.e. around F290, of the VSD. This helix conformation switch would remove the need for a physical rotation of the entire S4 $\alpha$-helix and minimize the energetics of the transitions.

In the years that followed the elucidation of the crystal structures of Kv1.2 and Kv1.2/2.1 chimera, both depicting the VSD in the activated state, there commenced intense research activity using multiple approaches aimed at understanding the position of the voltage sensor in the resting state and its interactions with the surroundings (Campos et al., 2007; DeCaen et al., 2008, 2009, 2011; Henrion et al., 2012; Pathak et al., 2007; Yarov-Yarovoy et al., 2012). Notable in this context are the long MD simulations made in Shaw's laboratory to visualize a complete translocation of a $K^+$ channel voltage sensor from the active state to the resting state upon membrane hyperpolarizations (Jensen et al., 2012).

## Other approaches to modelling channel gating

Molecular dynamics simulation represents a powerful approach to investigate, at atomic resolution, the conformational transitions occurring in channel gating. It is, however, still computationally very expensive, and simulation time scales present a major challenge. Molecular dynamics simulation also falls short on the grounds of model validation, i.e. the assessment of the extent to which its output reproduces the experimental results. To overcome these limitations, alternative strategies have emerged over the years that simplify the system studied by applying sensible approximations, without impacting the prospect of obtaining basic information on the dynamics of the gating structures. Importantly, these approaches also provide models that can be validated.

**Alternative approaches to investigate voltage sensor gating.** We recall here the mesoscale model of Peyser and Nonner, wherein the voltage sensor was represented by point charges immersed in a homogeneous dielectric environment, and statistical mechanics were applied to determine the stability of the various gating states (Peyser & Nonner, 2012a, 2012b). Model behaviour appeared very robust and highly predictive of experimental data. Unfortunately, the model would operate only at equilibrium, and thus was unable to predict the dynamic features of macroscopic gating currents.

Another macroscopic approach to describe voltage gating was proposed by Wharshel's group (Dryga et al., 2012a, 2012b; Kim & Warshel, 2014). They used meta-dynamics-based algorithms to find a reasonable energy profile for the movement of the voltage sensor

## Box 5. The five-state gating model

K$^+$ channel deactivation has been shown to encompass five stable states of the VSD, with the voltage sensor moving inwards by 10−15 Å, and the gating charges on S4 sequentially engaging in forming and breaking ion pairs with acidic residues of the VSD external vestibule and with the gating charge transfer centre (GCTC), comprising the phenylalanine F233 and the two acidic residues on S2 and S3 (Figs B5A and B5B).

during activation. With the energy profile known, they predicted the dynamics of the voltage sensor using the Langevin equation, and the gating currents from the voltage sensor movement and the voltage profile across the VSD. The model was able to predict several essential features of the gating current observed experimentally, such as the fast gating component and the rising phase present at the beginning of the depolarizing pulse at relatively high membrane potentials.

Bezanilla and co-workers recently proposed another model of voltage gating that assesses self-consistently, using a Poisson–Nernst–Planck formalism, the electrostatic energy resulting from the combination of the gating charges and the applied voltage (Horng et al., 2019). The gating charges, modelled as charged particles connected

to the S4 segment by springs, would move by electrodiffusion and provide the force for pulling S4 across the VSD. This model is also capable of reproducing the main features of the experimental gating current. Shortcomings of the model are that it does not include the fixed charges present in the VSD (that represent the counter-charges for the gating charges on the S4 segment and regulate its movement), and the gating charge distribution and the geometry of the voltage sensor were not derived from available crystal structures.

Our laboratory has also developed a model of voltage gating, based on both the Poisson–Nernst–Planck formalism for the description of ion electrodiffusion and the assessment of the electrostatic potential, and the Brownian dynamics for the description of the voltage

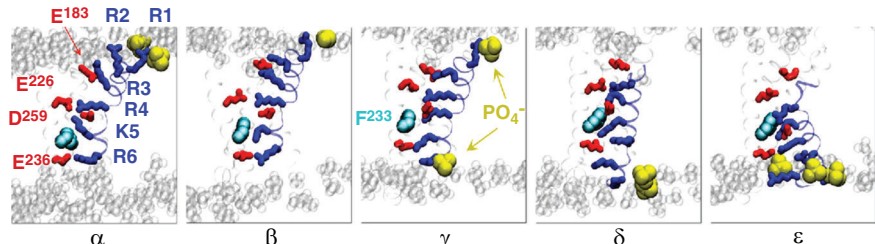

**Figure B5A.  The five-state gating model of Delemotte et al. (2011)**
Representative conformations ($\alpha$, $\beta$, $\gamma$, $\delta$ and $\varepsilon$) of the voltage-sensing domains resulting from molecular dynamics simulations showing the interactions of the S4 basic residues (blue sticks: R1, R2, R3, R4, K5 and R6) with their privileged binding sites (red sticks: E183, E226, D259 and E236) and with the lipid head group (yellow spheres, PO$_4$$^-$). The highly conserved residue F233 of S2 is shown as cyan spheres. [From Delemotte et al. (2011).]

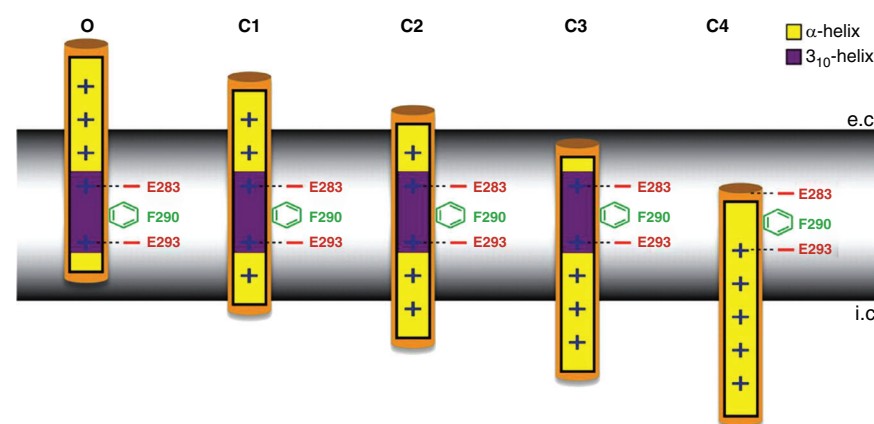

**Figure B5B. The five-state gating model of Henrion et al. (2012)**
Overlay of simulated voltage-sensing domain structures of the Kv1.2 channel after imposing harmonic restraints. Cartoon of the molecular models of voltage-sensing domain states during the gating process. [From Henrion et al. (2012).]

sensor movements (Catacuzzeno & Franciolini, 2019; Catacuzzeno et al., 2020). The geometry and charge distribution of the model were derived from a *Shaker* atomic structure obtained by homology modelling from the crystallographic structure of the Kv1.2/2.1 chimera (Long et al., 2007). The model agrees fully with recent MD and structural data indicating that during activation, the voltage sensor visits five states (cf. Delemotte et al., 2011; Henrion et al., 2012) and translocates during its motion the first four gating charges across the gating pore (Fig. 6*A*).

As a result, the energy profile seen by the voltage sensor during activation, assessed self-consistently with the Poisson equation, displays four barriers (for the four gating charges on the voltage sensor relevant in the gating process; Fig. 6*B*). The simulated gating currents obtained with our Brownian model reproduced all the main features of the gating currents recorded from typical $K^+$ channels (Catacuzzeno, Franciolini et al., 2021; Catacuzzeno, Sforna et al., 2021; Fig. 6*C*).

Our Brownian model was also able to reproduce accurately the gating current fluctuations (noise) from which individual charge packages transported by the voltage sensor during activation can be assessed. This is crucial for reinterpreting the classical experimental data on gating current noise that indicated shot current charge packages of $\sim 2.3e_0$ (Conti & Stühmer, 1989; Sigg et al., 1994), when recent gating models consistently suggest charge packages of $1.0e_0$. Our Brownian model has helped to resolve the conflict by showing that the relatively high charge shot deduced from the fluctuation analysis of experimental results from the limited recording bandwidth that makes the sequential gating charges crossing the gating pore in rapid succession to become indistinguishable individually, and thus appear as a single larger charge (Catacuzzeno, Franciolini et al., 2021; see also Crouzy & Sigworth, 1993).

## Conclusions and outlook

In their landmark papers describing the ionic basis of the action potential, Hodgkin and Huxley predicted the charged gating particles that needed to move across the membrane to account for the voltage-dependent $Na^+$ and $K^+$ permeability changes found in the squid giant axon. The predicted gating currents were observed some 20 years later (Armstrong & Bezanilla, 1973; Keynes & Rojas, 1974; Schneider & Chandler, 1973) and estimated to result from the outward movement of $12-14e_0$ across the electric field to gate a single $Na^+$ channel. Gating currents from $K^+$ channels were recorded later and found to share essentially the same properties as $Na^+$ gating currents. When the primary sequences of voltage-gated $Na^+$ and $K^+$ channels were elucidated, in the early 1980s, and their architecture was deduced, the fourth transmembrane domain, the S4 segment, was found systematically to contain an unexpectedly high number of positively charged residues, mainly arginine, and thus proposed to be the voltage sensor of the channels (the structural counterpart of the charged gating particles of Hodgkin and Huxley).

The first model of channel gating appeared in the form of the sliding helix, which postulated the outward, rotational movement of the putative voltage sensor, the S4 segment, with the gating charges sequentially forming ion pairs with the surroundings, to make the process energetically viable (Catterall, 1986; Guy & Seetharamulu, 1986). The first reliable high-resolution structures of crystallized voltage-gated ion channels (the Kv1.2, soon followed by the Kv1.2/2.1 chimera; Long et al., 2005a, 2005b, 2007) were both in the open state, with the S4 segment projected outwards, as expected from transmembrane voltage zeroing of the crystallization procedure. These atomic-resolution structures of voltage-gated channels allowed MD modelling to step in and contribute greatly to our understanding of voltage-dependent gating and to picture the resting

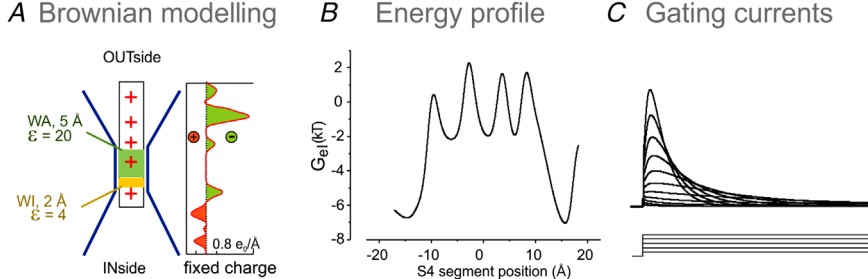

**Figure 6. Brownian model of channel gating**
*A*, schematic diagrams showing the geometry of the voltage-sensing domain of our model and the charges present in the S4 segment and in the rest of the voltage-sensing domain. *B*, energy profile experienced by the voltage sensor during its movement, at 0 mV. The energy profile is self-consistently assessed with the Poisson equation, considering the effect of all the charges present in the system. *C*, family of simulated gating currents obtained in response to depolarizing pulses. [From Catacuzzeno, Sforna et al., 2021.]

state of the channels. Brownian modelling simplified the systems studied by introducing sensible approximations as a trade-off for obtaining the much-needed macroscopic gating currents and their relevant parameters.

**Outlook.** Turning to future challenges in the field, the first goal is to obtain high-resolution cryo-EM or X-ray crystallographic structures of each relevant functional and structural state of the channel. Molecular dynamics and other modelling approaches have so far tried to compensate for this limitation by picturing the unseen (intermediate) conformational states. Now we need to have more direct evidence and knowledge of them, which is an absolute requirement to visualize the rearrangements of VSD between the various gating states at the atomic level. This would also permit resolution of the laws of physics that guide these rearrangements and attempt to provide a mechanistic interpretation of the full gating process. The main challenge in this context is to develop strategies to trap the VSD in the various gating states, in addition to structural techniques to resolve, for each of them, the position of atoms at the sub-ångström level.

A second point is to reinstate the importance of energetics in channel gating, which is much undervalued at present. The complex trajectories of the gating charge translocations among the various gating states follow predefined paths determined by the energy landscapes encountered. These paths might even be different for activation and deactivation, with the elected pathway depending on the boundary conditions used (Villalba-Galea & Chiem, 2020). *Shaker* and hERG channels are pertinent examples of this occurrence. Knowledge of these pathways and charting the energy profile would greatly help our understanding of the physical and molecular determinants of gating. Unfortunately, they are poorly known at present, because they are shaped by the structural conformations of the VSD in its various states. Future studies should start trying to interpret data through VSD models that strictly correlate function and structure. Some of these models have begun to appear, which assess the electrostatic energy landscape seen by the voltage sensor using Poisson's equation and considering all the charges present in the three-dimensional structure of the *Shaker* VSD (Catacuzzeno & Franciolini, 2019).

Thirdly, we would encourage expansion of the use of multiscale approaches in the study of channel gating. We are aware of the limits of MD modelling with regard to the duration of simulations and output verification. Brownian models overcome these limits, yet they are not immune from other limits, for instance determining crucial parameters needed in their own modelling. To unite the forces of the two models, in a recent study we used a multiscale hierarchical approach to test whether $Na^+$ and $K^+$ channels respond with different activation rates to potential change mainly because of the different polarizability in the gating pore (determined by whether there is a threonine or an isoleucine in the channel at the equivalent position of 287 of *Shaker*). By first performing all-atom MD simulations on the atomic structure of *Shaker* VSD to assess the effective dielectric constant in the VSD of both the wild-type and the *in silico* mutated I287 with threonine (I287T), as in the $Na^+$ channel, and then using these MD computations to assess the polarizability of the environment surrounding the voltage sensor needed for application of our Brownian model, we were able to reproduce the effects of the isoleucine-to-threonine mutation of residue 287 on the time course of the gating currents and, in turn, on the firing threshold (Catacuzzeno & Franciolini, 2019; Catacuzzeno, Sforna et al., 2021).

Finally, we think that more consideration should be given to electrodiffusive theories in modelling channel gating. Most experimental data on voltage-dependent gating accumulated over the past 50 years have been interpreted within the framework of rate models, an approach pioneered by Hodgkin and Huxley in their quantitative model of the gating charge movement during channel activation. Although this approach has undoubtedly been very useful to understand many key points about voltage-dependent gating, it is now clear that its application to channel gating might not always be appropriate or the best choice. The structural and functional information accumulated over the years shows that voltage-dependent gating, namely the movement of the S4 segment across the membrane electric field, has many features of an electrodiffusive process. It can be approximated by a rate model only if the diffusive particles experience an energy profile characterized by a relatively high (i.e. >4–5 kT) energy barrier separating the stable states. Our Brownian model of voltage gating, in addition to information from MD simulations, tells us that this condition might not always be present in real channels. In addition, rate models require only the rate constants as parameters connecting the different states; hence, they are not able to tell us much about the structural basis of the voltage-dependent gating. Thus, expanding the usage of electrodiffusive models wherein the three-dimensional structure of the VSD is more explicitly considered will certainly be most welcome.

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

## Additional Information

### Competing interests

The authors declare no competing financial interests.

### Author contributions

Both authors carried out the literature search and together wrote and revised the manuscript.

### Funding

Our original research has been supported over the years by grants from Cassa di Risparmio di Perugia.

### Acknowledgements

We are grateful to Bob Eisenberg and Pancho Bezanilla for the several discussions we had on the subject, and to Sandy Harper for reading and commenting on drafts of the manuscript.

Open Access Funding provided by Universita degli Studi di Perugia within the CRUI-CARE Agreement.

### Keywords

channel gating, gating currents, gating models, Hodgkin and Huxley model, ion channels

### Supporting information

Additional supporting information can be found online in the Supporting Information section at the end of the HTML view of the article. Supporting information files available:

**Peer Review History**

