## [Peer Review History · The Journal of Physiology]

Due 31 January: Topical Review: The 70-year search of the voltage sensor of ion channels

Fabio Franciolini and Luigi Catacuzzeno
DOI: 10.1113/JP282780

Corresponding author(s): Luigi Catacuzzeno (luigi.catacuzzeno@unipg.it)

The following individual(s) involved in review of this submission have agreed to reveal their identity: Fredrik Elinder (Referee #1); James Letts (Referee #3)

Review Timeline:

Submission Date:	27-Jan-2022
Editorial Decision:	09-Mar-2022
Revision Received:	06-Apr-2022
Accepted:	25-Apr-2022

Senior Editor: Ian Forsythe

Reviewing Editor: Thomas DeCoursey

Transaction Report:

Dear Dr Catacuzzeno,

Re: JP-TR-2022-282780 "Due 31 January: Topical Review: The 70-year search of the voltage sensor of ion channels" by Fabio Franciolini and Luigi Catacuzzeno

Thank you for submitting your Topical Review to The Journal of Physiology. It has been assessed by a Reviewing Editor and by 3 expert referees and I am pleased to tell you that it is considered to be acceptable for publication following satisfactory revision.

The reports are copied at the end of this email. Please address all of the points and incorporate all requested revisions, or explain in your Response to Referees why a change has not been made.

NEW POLICY: In order to improve the transparency of its peer review process The Journal of Physiology publishes online as supporting information the peer review history of all articles accepted for publication. Readers will have access to decision letters, including all Editors' comments and referee reports, for each version of the manuscript and any author responses to peer review comments. Referees can decide whether or not they wish to be named on the peer review history document.

I hope you will find the comments helpful and have no difficulty in revising your manuscript within 4 weeks.

Your revised manuscript should be submitted online using the links in Author Tasks Link Not Available. This link is to the Corresponding Author's own account, if this will cause any problems when submitting the revised version please contact us.

You should upload:

- A Word file of the complete text (including any Tables);
- An Abstract Figure, (with accompanying Legend in the article file)
- Each figure as a separate, high quality, file;
- A full Response to Referees;
- A copy of the manuscript with the changes highlighted.
- Author profile. A short biography (no more than 100 words for one author or 150 words in total for two authors) and a portrait photograph of the two leading authors on the paper. These should be uploaded, clearly labelled, with the manuscript submission. Any standard image format for the photograph is acceptable, but the resolution should be at least 300 dpi and preferably more.

- A 'Cover Art' file for consideration as the Issue's cover image;
- Appropriate Supporting Information (Video, audio or data set https://jp.msubmit.net/cgi-bin/main.plex?form_type=display_requirements#supp).

To create your 'Response to Referees' copy all the reports, including any comments from the Senior and Reviewing Editors into a Word, or similar, file and respond to each point in colour or CAPITALS. Upload this when you submit your revision.

I look forward to receiving your revised submission.

Yours sincerely,

Ian D. Forsythe
Deputy Editor-in-Chief
The Journal of Physiology
<https://jp.msubmit.net>
<http://jp.physoc.org>
The Physiological Society
Hodgkin Huxley House
30 Farringdon Lane
London, EC1R 3AW
UK
<http://www.physoc.org>
<http://journals.physoc.org>

EDITOR COMMENTS

Reviewing Editor:

The reviewers have provided a number of excellent and constructive suggestions for improving this review. Please consider them in the revision. In addition I have a few suggestions:

1) The title is good, but perhaps misleading. The true search is for the voltage-sensing mechanism, not the physical voltage sensor per se! Structure is important, but does not necessarily reveal mechanism, which is what we really want to know. Perhaps alter the title to "The 70-year search for the voltage sensing mechanism of ion channels."

2) The title implies that the review covers all (voltage-gated) ion channels. This is not the case. Despite the similarity of their structures and mechanisms to those of K and Na channels, Ca and H channels are not discussed. This is fine, but either the title should be restricted to Na and K channels, or there should be a sentence in the manuscript explaining that despite the similar voltage-gating mechanisms of other channels, the review is limited to the two channels that Hodgkin & Huxley studied.

3) Perhaps in the Introduction, or early on, it might be instructive to mention two reasons Hodgkin & Huxley succeeded. First, they chose an ideal preparation, the enormous squid axon that could be dissected easily and voltage-clamped using a simple axial wire. Second, they had the brilliance (or serendipity) to elect to control the membrane potential and measure membrane current, rather than to control the current and measure voltage. The fact that voltage-gated ion channels are in fact controlled by voltage means that membrane potential should be the independent variable. If you wonder whether this choice made interpretation of the data more straightforward and intelligible, just take a look at impedance diagrams!

Senior Editor:

Thank you for an interesting review. After responding to the referees the authors need to re-write the abstract in the JP style, which should contain concise factual content about the topic under review and come to a clear conclusion in the last sentence (Please avoid writing the abstract as an outline, but make your findings explicit).

REFeree COMMENTS

Referee #1:

This review by Franciolini and Catacuzzeno about "The 70-year search of the voltage sensor of ion channels" is insightful and very well written, and, to my knowledge, it correctly describes the historically important steps.

I have only a few minor comments.

Page 1: I suggest to include the first paper of the 1952 series of papers by Hodgkin and Huxley, the one together with Bernard Katz.

Page 1: "Loligo" should be italicized.

Page 1: condstucted should be constructed

Page 1: Is the reference to Moore (1967) relevant? I cannot see that this paper is mentioning ions passing across the membrane at all.

Page 2: "..., and with much faster time course than the rising phase (Figure 1)". I do not understand this. First of all, the two time courses in Fig.1 have similar time constants, thus not "much faster". Secondly, the time courses depend on the chosen voltage for opening and closing. This should be clarified.

Page 4: "On the contrary there was instead a high level of cooperativity between the four voltage sensors in the final step that would open the channel." I guess a reference is missing here. The reference in the previous sentence, to Taylor & Bezanilla (1983), does not describe this as far as I know.

Page 10, Box 2: Fig B2 is said to be "homemade". I guess this expression can be removed if the authors have done it.

Referee #2:

The manuscript by Franciolini and Catacuzzeno contains a historical recount of our current understanding of voltage-sensing in ion channels. The review summarizes well landmark events, starting with the publication of the seminal work by Hodgkin and Huxley and ending with the elucidations of structural models for voltage-gated ion channels. Although the outline of the manuscript is well thought out, the writing requires extensive revision. In addition, the authors link the activity of voltage sensors with the structural information available, discussing both the advantage and limitations of this exercise. This is timely and urgently needed in the field given the current pour of structures in the ion channel field. Yet, the authors fell short in providing the reader with a more comprehensive discussion of the danger of inappropriate overinterpretation of the structural data. There are concerns that must be addressed before the publication of the manuscripts. Many of these concerns are in form, as the grammar needs to be revised. Other concerns are conceptual. Revising these latter ones would provide the reader with a complete panoramic of the field's current status. Those concerns are as follows:

Conceptual concerns:

Page 2, third paragraph: the authors highlight that the value of "n" found by Hodgkin and Huxley was 4. This is a remarkable prediction that passed the test of time because it stated that 4 independent voltage sensors are controlling the activity of the K⁺-conductance. The authors should emphasize this remarkable finding as an extraordinary one because it correlates with the four subunits forming a canonical Kv channel.

Page 3, second paragraph: Like before, the authors should link the prediction from HH of m=3 with what we know today. Domains I through III seems to dominate the activation process in Nav channels, while Domain IV modulates inactivation.

Page 4, second paragraph: The authors make the statement "On the contrary there was instead a high level of cooperativity between the four voltage sensors in the final step that would open the channel." There are several references missing here and those needs to be added as the "concerted step" is a critical process in channel activation and has been extensively studied, with the underlying mechanism remaining unclear.

Page 5, first paragraph: For the equation $pO = \{1/(1 + \exp[-z_g \cdot e \cdot (V - V_{1/2})/kBT])\}^4$, the authors should make clear to the reader that it is important to understand that this equation represents 4 parallel two-state models for the voltage sensor controlling the opening of a channel.

Page 5, second paragraph: When the authors states that "...which provided values of 6 and 4.5 elementary charges, e₀, ...", it is important to show the readers that the reason the number of charges is underestimate because the underlying model justifying the equation consists of for two-state voltage sensors. The fact that it is a two-state model causes the charge to be

underestimated. Example of how a two-state model results in that was provided by Bezanilla and Villalba-Galea (2013, JGP).

Page 7, box 1: It should be clarified that this is a modernized version of the original sliding helix model.

Page 14, third paragraph: I believe the authors' conclusion has a short scope, with a lot of room for improvement. However, in addition to what it is stated, the authors should add that even in the case in which the native environment of channels was conserved, there are thermodynamical considerations to be made as well when it comes to understand what the structural model in question represents. For instance, a recent structure of the human Kv7.2 associated with calmodulin (PDB: 7CR3) shows activated voltage sensors with a likely closed pore for a channel that does not inactivate. On another example, the VSD of channels undergoes voltage-independent transitions following voltage-dependent ones. Evidence for such transitions have been well documented yet largely ignored since the 1980s in Nav channels, Shaker, HCN, hERG and other channels (Villalba-Galea and Chiem, 2020, *Frontiers*). Furthermore, the existence of such transitions implies that the "open" state must likely be meta-stable and hardly attainable with the current approaches employed in structural biology. In agreement with the authors, the previous examples lead to the inexorable conclusion that channels adopt conformations beyond the simple opening and closing of their pores, and structural data offer a wealth of information that is, at times, hard to interpret.

Page 22, third paragraph: As part of their second point, the authors should highlight that the paths for activation and deactivation are likely not the mutual reverse process. Instead, they follow unique steps that can vary with activity. Examples like Shaker, Kv3.1, and hERG should be considered as instances of this topic.

Referee #3:

In their retrospective entitled "The 70-year search of voltage-dependent gating models in ion channels" Catacuzzeno and Franciolini review the progress on our understanding of the mechanism of voltage-gating since the groundbreaking work of Hodgkin and Huxley. Overall the manuscript is very well written and presents a clear historical narrative of the progress in this field.

Comments

The figure labeling and referencing needs to be made consistent throughout.

There is a reference to Figure 1 on page 2 but there is no Figure 1. I believe this should be Figure 2.1.

The authors refer to Figure 3.1B in the text but never to Figure 3.1A.

The authors are inconsistent with their references to figures in the Boxes, for example, they refer to Figure B1 in Box 1 but there are no references to Figure B2 in Box 2 (same for Figures B5A and B5B in Box 5).

There is a reference to Figure 5.3 in Box 4 but no reference to Figure B4.

There is no reference to Figure 6.1 in the text.

The referencing to Figure 7.1A, Figure 7.1B and Figure 7.1C on page 20 are out of order (A, C then B). Either the order of the figure should be altered or the discussion in the text should be changed.

The text on pages 12-13 needs to be better written to accommodate Box 4. It appears that the authors had written the text as a single section and then split some of it into Box 4 without altering the main text (this can be seen also from the Figure 5.2 reference error above). This results in the main text losing logical consistency without the reader having to first read Box 4 as it is presented in the pdf. This needs to be corrected. Either remove Box 4 and just keep it all as one continuous section (as it appears to have been originally written) or rewrite the main text so that it flows consistently without being dependent on the text in Box 4.

Page 2, paragraph 2, sentence 1: there is an extra comma that needs to be removed

Page 8, section 4.2, paragraph 1, sentence 1: "cysteine binding compounds, as menthanethiosulfonate (MTS)..." should be "cysteine binding compounds, such as menthanethiosulfonate (MTS)..."

BOX 2, sentence 1: "To accommodate its FRET results..." should be "To accommodate their FRET results..."

Page 20, paragraph 5, sentence 1: "As result..." should be "As a result..."

Page 22, the final sentence before "Outlook" section: "Brownian modeling, which simplified the systems studied by introducing sensible approximations as a tradeoff for creating the much needed macroscopic gating currents and their relevant parameters." is a sentence fragment, it is missing a verb, what did the Brownian modeling do? It is unclear but maybe this is what was meant ""Brownian modeling simplified the systems studied by introducing sensible approximations as a tradeoff for creating the much-needed macroscopic gating currents and their relevant parameters."

REQUIRED ITEMS:

-Please include an Abstract Figure. The Abstract Figure is a piece of artwork designed to give readers an immediate understanding of the Review Article and should summarise the main conclusions. If possible, the image should be easily 'readable' from left to right or top to bottom. It should show the physiological relevance of the Review so readers can assess the importance and content of the article. Abstract Figures should not merely recapitulate other figures in the Review. Please try to keep the diagram as simple as possible and without superfluous information that may distract from the main conclusion of the Review. Abstract Figures must be provided by authors no later than the revised manuscript stage and should be uploaded as a separate file during online submission labelled as File Type 'Abstract Figure'. Please ensure that you include the figure legend in the main article file. All Abstract Figures will be sent to a professional illustrator for redrawing and you may be asked to approve the redrawn figure before your paper is accepted.

-Your MS must include a complete "Additional information section" with the following 4 headings and content:

Competing Interests: A statement regarding competing interests. If there are no competing interests, a statement to this effect must be included. All authors should disclose any conflict of interest in accordance with journal policy.

Author contributions: Each author should take responsibility for a particular section of the study and have contributed to writing the paper. Acquisition of funding, administrative support or the collection of data alone does not justify authorship;

these contributions to the study should be listed in the Acknowledgements. Additional information such as 'X and Y have contributed equally to this work' may be added as a footnote on the title page.

It must be stated that all authors approved the final version of the manuscript and that all persons designated as authors qualify for authorship, and all those who qualify for authorship are listed.

Funding: Authors must indicate all sources of funding, including grant numbers. If authors have not received funding, this must be stated.

It is the responsibility of authors funded by RCUK to adhere to their policy regarding funding sources and underlying research material. The policy requires funding information to be included within the acknowledgement section of a paper. Guidance on how to acknowledge funding information is provided by the Research Information Network. The policy also requires all research papers, if applicable, to include a statement on how any underlying research materials, such as data, samples or models, can be accessed. However, the policy does not require that the data must be made open. If there are considered to be good or compelling reasons to protect access to the data, for example commercial confidentiality or legitimate sensitivities around data derived from potentially identifiable human participants, these should be included in the statement.

Acknowledgements: Acknowledgements should be the minimum consistent with courtesy. The wording of acknowledgements of scientific assistance or advice must have been seen and approved by the persons concerned. This section should not include details of funding.

-Please upload separate high quality figure files via the submission form.

-Author profile(s) must be uploaded via the submission form. Authors should submit a short biography (no more than 100 words for one author or 150 words in total for two authors) and a portrait photograph of the two leading authors on the paper. These should be uploaded, clearly labelled, with the manuscript submission. Any standard image format for the photograph is acceptable, but the resolution should be at least 300 dpi and preferably more. A group photograph of all authors is also acceptable, providing the biography for the whole group does not exceed 150 words.

-It is the authors' responsibility to obtain any necessary permissions to reproduce previously published material
https://jp.msubmit.net/cgi-bin/main.plex?form_type=display_requirements#use

END OF COMMENTS

Confidential Review

27-Jan-2022

The manuscript by Franciolini and Catacuzzeno contains a historical recount of our current understanding of voltage-sensing in ion channels. The review summarizes well landmark events, starting with the publication of the seminal work by Hodgkin and Huxley and ending with the elucidations of structural models for voltage-gated ion channels. Although the outline of the manuscript is well thought out, the writing requires extensive revision. In addition, the authors link the activity of voltage sensors with the structural information available, discussing both the advantage and limitations of this exercise. This is timely and urgently needed in the field given the current pour of structures in the ion channel field. Yet, the authors fell short in providing the reader with a more comprehensive discussion of the danger of inappropriate overinterpretation of the structural data. There are concerns that must be addressed before the publication of the manuscripts. Many of these concerns are in form, as the grammar needs to be revised. Other concerns are conceptual. Revising these latter ones would provide the reader with a complete panoramic of the field's current status. Those concerns are as follows:

Conceptual concerns:

Page 2, third paragraph: the authors highlight that the value of "n" found by Hodgkin and Huxley was 4. This is a remarkable prediction that passed the test of time because it stated that 4 independent voltage sensors are controlling the activity of the K⁺-conductance. The authors should emphasize this remarkable finding as an extraordinary one because it correlates with the four subunits forming a canonical Kv channel.

Page 3, second paragraph: Like before, the authors should link the prediction from HH of m=3 with what we know today. Domains I through III seems to dominate the activation process in Nav channels, while Domain IV modulates inactivation.

Page 4, second paragraph: The authors make the statement "*On the contrary there was instead a high level of cooperativity between the four voltage sensors in the final step that would open the channel.*" There are several references missing here and those needs to be added as the "concerted step" is a critical process in channel activation and has been extensively studied, with the underlying mechanism remaining unclear.

Page 5, first paragraph: For the equation $p_o = \{1/(1+ \exp[-z_g.e.(V-V_{1/2})/kBT])\}^4$, the authors should make clear to the reader that it is important to understand that this equation represents 4 parallel two-state models for the voltage sensor controlling the opening of a channel.

Page 5, second paragraph: When the authors states that "*...which provided values of 6 and 4.5 elementary charges, e0, ...*", it is important to show the readers that the reason the number of charges is underestimate because the underlying model justifying the equation consists of for two-state voltage

sensors. The fact that it is a two-state model causes the charge to be underestimated. Example of how a two-state model results in that was provided by Bezanilla and Villalba-Galea (2013, JGP).

Page 7, box 1: It should be clarified that this is a modernized version of the original sliding helix model.

Page 14, third paragraph: I believe the authors' conclusion has a short scope, with a lot of room for improvement. However, in addition to what is stated, the authors should add that even in the case in which the native environment of channels was conserved, there are thermodynamical considerations to be made as well when it comes to understand what the structural model in question represents. For instance, a recent structure of the human Kv7.2 associated with calmodulin (PDB: 7CR3) shows activated voltage sensors with a likely close pore for a channel that does not inactivate. On another example, the VSD of channels undergoes voltage-independent transitions following voltage-dependent ones. Evidence for such transitions has been well documented yet largely ignored since the 1980s in Nav channels, Shaker, HCN, hERG and other channels (Villalba-Galea and Chiem, 2020, *Frontiers*). Furthermore, the existence of such transitions implies that the "open" state must likely be meta-stable and hardly attainable with the current approaches employed in structural biology. In agreement with the authors, the previous examples lead to the inexorable conclusion that channels adopt conformations beyond the simple opening and closing of their pores, and structural data offer a wealth of information that is, at times, hard to interpret.

Page 22, third paragraph: As part of their second point, the authors should highlight that the paths for activation and deactivation are likely not the mutual reverse process. Instead, they follow unique steps that can vary with activity. Examples like Shaker, Kv3.1, and hERG should be considered as instances of this topic.

Other concerns:

There are several instances in which the authors have 4- or 6-line sentences that are hard to follow. The authors should also revise the manuscript for grammar. Some examples are:

Page 1 last paragraph: "How and where these currents crossed the membrane at that time remained unresolved" and "In their papers Hodgkin and Huxley referred only..."

Page 2, end of first paragraph: "ms" should be spelled out

Page 2, second paragraph: "Looking first at the time dependence of g_K , which is easier to describe, on applying a depolarizing step it rises slowly following a sigmoidal time course, , until it reaches a steady level"

Page 5, third paragraph: the first sentence must be revised as the subject of the sentence is between parentheses.

From: jp@physoc.org <jp@physoc.org>
Sent: Wednesday, March 9, 2022 16:40
To: Luigi Catacuzzeno <luigi.catacuzzeno@unipg.it>
Subject: JP-TR-2022-282780 - The Journal of Physiology - Decision Letter

Dear Dr Catacuzzeno,

Re: JP-TR-2022-282780 "Due 31 January: Topical Review:

The 70-year search of the voltage sensor of ion channels" by Fabio Franciolini and Luigi Catacuzzeno

Thank you for submitting your Topical Review to The Journal of Physiology. It has been assessed by a Reviewing Editor and by 3 expert referees and I am pleased to tell you that it is considered to be acceptable for publication following satisfactory revision.

The reports are copied at the end of this email. Please address all of the points and incorporate all requested revisions, or explain in your Response to Referees why a change has not been made.

I hope you will find the comments helpful and have no difficulty in revising your manuscript within 4 weeks.

You should upload:

- A Word file of the complete text (including any Tables);
- An Abstract Figure (with accompanying Legend in the article file)
- Each figure as a separate, high quality, file;
- A full Response to Referees*;
- A copy of the manuscript with the changes highlighted.
- Author profile. A short biography (no more than 100 words for one author or 150 words in total for two authors) and a portrait photograph of the two leading authors on the paper. These should be uploaded, clearly labelled, with the manuscript submission. Any standard image format for the photograph is acceptable, but the resolution should be at least 300 dpi and preferably more.

* To create your 'Response to Referees' copy all the reports, including any comments from the Senior and Reviewing Editors into a Word, or similar, file and respond to each point in colour or CAPITALS. Upload this when you submit your revision.

I look forward to receiving your revised submission.

Yours sincerely,

Ian D. Forsythe
Deputy Editor-in-Chief of The Journal of Physiology

EDITOR COMMENTS

Reviewing Editor:

The reviewers have provided a number of excellent and constructive suggestions for improving this review. Please consider them in the revision. In addition I have a few suggestions:

1) The title is good, but perhaps misleading. The true search is for the voltage-sensing mechanism, not the physical voltage sensor per se! Structure is important, but does not necessarily reveal mechanism, which is what we really want to know. Perhaps alter the title to "**The 70-year search for the voltage sensing mechanism of ion channels.**"

Suggestion adopted. The title now reads "The 70-year search for the voltage sensing mechanism of ion channels"

2) The title implies that the review covers all (voltage-gated) ion channels. This is not the case. Despite the similarity of their structures and mechanisms to those of K and Na channels, Ca and H channels are not discussed. This is fine, but either the title should be restricted to Na and K channels, or there should be a sentence in the manuscript explaining that despite the similar voltage-gating mechanisms of other channels, the review is limited to the two channels that Hodgkin & Huxley studied.

Suggestion taken. At p. 4 we inserted the following sentence: "We add that despite similar voltage gating mechanisms are shared by other channels, the scope of this review is restricted to the two channels (actually conductances) studied by Hodgkin and Huxley".

3) Perhaps in the Introduction, or early on, it might be instructive to mention two reasons Hodgkin & Huxley succeeded. First, they chose an ideal preparation, the enormous squid axon that could be dissected easily and voltage-clamped using a simple axial wire. Second, they had the brilliance (or serendipity) to elect to control the membrane potential and measure membrane current, rather than to control the current and measure voltage. The fact that voltage-gated ion channels are in fact controlled by voltage means that membrane potential should be the independent variable. If you wonder whether this choice made interpretation of the data more straightforward and intelligible, just take a look at impedance diagrams!

Suggestion accepted. At p. 5 we inserted the following sentence: "The following reasons turned out to be critical for Hodgkin and Huxley to succeed in their endeavour to model the action potential waveform and describe the underlying currents. First, was the choice of the squid giant axon that could be voltage-clamped effectively using a simple axial wire electrode. Second, they elected to control the membrane potential and measure membrane current on the grounds that membrane conductances to ions were controlled by voltage. Third, that the squid giant axon is extremely minimalist in terms of ion currents, displaying essentially only the Na and the K currents they studied".

Senior Editor:

Thank you for an interesting review. After responding to the referees the authors need to re-write the abstract in the JP style, which should contain concise factual content about the topic under review and come to a clear conclusion in the last sentence (Please avoid writing the abstract as an outline, but make your findings explicit).

Abstract

This retrospective on the voltage sensing mechanisms and gating models of ion channels begins in 1952 with the 'charged gating particles' postulated by Hodgkin and Huxley, and viewed as charges moving across the membrane and controlling its permeability to Na and K ions. Hodgkin and Huxley postulated that their movement should generate small and fast capacitive currents, that were recorded 20 years later as gating currents. In the early 1980s several voltage-gated channels were cloned, and found to possess a common architecture: four homologous domains or subunits, each displaying six transmembrane α -helical segments, with the fourth segment (S4) displaying 4 to 7 positive charges invariably separated by two non-charged residues. This immediately suggested that this segment was serving as the channels' voltage sensor (the molecular counterpart of the charged gating particle postulated by Hodgkin and Huxley), and led to the development of the 'sliding helix' model. Twenty years later the X-ray crystallographic structures of many voltage-gated channels allowed investigation of their gating by molecular dynamics. Further understanding on channels gating will need the acquisition of high resolution structures of each of their relevant functional or structural states. This will allow the application of molecular dynamics and other approaches. Key will also be investigation of the energetics of channel gating, permitting understanding the physical and molecular determinants of gating. The use of multi-scale hierarchical approaches may well prove to be a rewarding strategy to overcoming the limits of the various single approaches to the study of channel gating.

REFEREE COMMENTS

Referee #1:

This review by Franciolini and Catacuzzeno about "The 70-year search of the voltage sensor of ion channels" is insightful and very well written, and, to my knowledge, it correctly describes the historically important steps.

I have only a few minor comments.

Page 1: I suggest to include the first paper of the 1952 series of papers by Hodgkin and Huxley, the one together with Bernard Katz.

Suggested reference inserted.

Page 1: "Loligo" should be italicized.

Fixed.

Page 1: condstucted should be constructed

Fixed.

Page 1: Is the reference to Moore (1967) relevant? I cannot see that this paper is mentioning ions passing across the membrane at all.

Reference to Moore 1967 has been removed.

Page 2: "..., and with much faster time course than the rising phase (Figure 1)". I do not understand this. First of all, the two time courses in Fig.1 have similar time constants, thus not "much faster". Secondly, the time courses depend on the chosen voltage for opening and closing. This should be clarified.

The reviewer's comments are pertinent. We have removed the incriminated part of the sentence, that is, "and with much faster time course than the rising phase".

Page 4: "On the contrary there was instead a high level of cooperativity between the four voltage sensors in the final step that would open the channel." I guess a reference is missing here. The reference in the previous sentence, to Taylor & Bezanilla (1983), does not describe this as far as I know.

We fixed it by inserting the following reference(s), "Zagotta et al., 1994; Smith-Maxwell et al., 1998; Pathak et al., 2005"

Page 10, Box 2: Fig B2 is said to be "homemade". I guess this expression can be removed if the authors have done it.

"Homemade" removed. Incidentally, it was just an internal memo.

Referee #2:

The manuscript by Franciolini and Catacuzzeno contains a historical recount of our current understanding of voltage-sensing in ion channels. The review summarizes well landmark events, starting with the publication of the seminal work by Hodgkin and Huxley and ending with the elucidations of structural models for voltage-gated ion channels. Although the outline of the manuscript is well thought out, the writing requires some revision. In addition, the authors link the activity of voltage sensors with the structural information available, discussing both the advantage and limitations of this exercise. This is timely and urgently needed in the field given the current pour of structures in the ion channel field. Yet, the authors fell short in providing the reader with a more comprehensive discussion of the danger of inappropriate overinterpretation of the structural data. There are concerns that must be addressed before the publication of the manuscripts. Many of these concerns are in form, as the grammar needs to be revised. Other concerns are conceptual. Revising these latter ones would provide the reader with a complete panoramic of the field's current status. Those concerns are as follows:

As for the concerns regarding form/grammar, the whole Ms has been reviewed by our colleague Sandy Harper, former lecturer in Physiology at the University of Aberdeen.

Conceptual concerns:

Page 2, third paragraph: the authors highlight that the value of "n" found by Hodgkin and Huxley was 4. This is a remarkable prediction that passed the test of time because it stated that 4 independent voltage sensors are controlling the activity of the K⁺-conductance. The authors should emphasize this remarkable finding as an extraordinary one because it correlates with the four subunits forming a canonical Kv channel.

Page 3, second paragraph: Like before, the authors should link the prediction from HH of m=3 with what we know today. Domains I through III seems to dominate the activation process in Nav channels, while Domain IV modulates inactivation.

Both suggestions (Page 2 and Page 3) taken. At p. 4 of revised Ms we inserted the following sentence: "The values of m and n found by Hodgkin and Huxley represent a remarkable prediction that has stood the test of time, to date. $n = 4$ for g_K correlates with the 4 subunits forming a typical Kv channel. So is the value of $m = 3$ for g_{Na} , as we now know that the first three domains – DI through DIII – arguably dominate the activation process in Nav channels, with domain IV apparently modulating inactivation."

Page 4, second paragraph: The authors make the statement "On the contrary there was instead a high level of cooperativity between the four voltage sensors in the final step that would open the channel." There are several references missing here and those needs to be added as the "concerted step" is a critical process in channel activation and has been extensively studied, with the underlying mechanism remaining unclear.

Missing references with regard to cooperativity has been raised also by reviewer 1. We believe we have addressed this comment by inserting the following references: Zagotta et al., 1994; Smith-Maxwell et al., 1998; Pathak et al., 2005 (cf. page 6).

Page 5, first paragraph: For the equation $p_o = \{1/(1 + \exp[-z_g \cdot e \cdot (V - V_{1/2})/kBT])\}^4$, the authors should make clear to the reader that it is important to understand that this equation represents 4 parallel two-state models for the voltage sensor controlling the opening of a channel.

The recommendation is helpful, thank you. We have replaced the sentence "(to the fourth power for a channel displaying four gating particles or voltage sensors)" with "(to the fourth power as the equation represents four parallel two-state models for the voltage sensor controlling the opening of a channel)" (cf. page 6).

Page 5, second paragraph: When the authors state that "...which provided values of 6 and 4.5 elementary charges, e_0 , ...", it is important to show the readers that the reason the number of charges is underestimated is because the underlying model justifying the equation consists of four two-state voltage sensors. The fact that it is a two-state model causes the charge to be underestimated. Example of how a two-state model results in that was provided by Bezanilla and Villalba-Galea (2013, JGP).

Suggestion followed. At the end of the paragraph we have added the following sentence: "The reasons why two-state models cause the translocated charge to be underestimated have been provided for instance by Bezanilla and Villalba-Galea (2013)" (cf. page 7).

Page 7, box 1: It should be clarified that this is a modernized version of the original sliding helix model.

We have now clarified it by replacing in the figure legend the sentence "Sketch of the Sliding helix model ..." with "Sketch of a modernized version of the original Sliding helix model ..." (cf. page 9).

Page 14, third paragraph: I believe the authors conclusion has a short scope, with a lot of room for improvement. However, in addition to what it is stated, the authors should add that even in the case in which the native environment of channels was conserved, there are thermodynamical considerations to be made as well when it comes to understand what the structural model in question represents. For instance, a recent structure of the human Kv7.2 associated with calmodulin (PDB: 7CR3) shows activated voltage sensors with a likely close pore for a channel that does not inactivate. On another example, the VSD of channels undergoes voltage-independent

transitions following voltage-dependent ones. Evidence for such transitions have been well documented yet largely ignored since the 1980s in Nav channels, Shaker, HCN, hERG and other channels (Villalba-Galea and Chiem, 2020, *Frontiers*). Furthermore, the existence of such transitions implies that the "open" state must likely is meta-stable and hardly attainable with the current approaches employed in structural biology. In agreement with the authors, the previous examples lead to the inexorable conclusion that channels adopt conformations beyond the simple opening and closing of their pores, and structural data offer a wealth of information that is, at time, hard to interpret.

We appreciate the reviewer's suggestion to improve our conclusions by replacing the last paragraph with the following expanded one. "In concluding this section we are mindful that structural data taken by themselves can often be misleading when translated directly into functional terms. Channels structures obtained by crystallization may only poorly reflect the actual structure of the native channels in the cell membrane (due to preparatory procedures for protein crystallization that rarely, if ever, preserve the natural biological environment). Even in the case in which the native environment of channels is conserved, there are thermodynamic considerations to be made as well when it comes to understanding what the structural model in question represents. For instance, a recent structure of the human Kv7.2 associated with calmodulin shows activated voltage sensors with a likely close pore for a channel that does not inactivate (Li *et al.*, 2021). Conversely, the hERG channel structure, found with an activated voltage sensor, displays an apparently conducting selectivity filter, in spite this channel undergoes a rapid inactivation at depolarized potentials (Wang & MacKinnon, 2017). Considering another example, the VSD of channels undergoes voltage-independent transitions following voltage-dependent ones (Villalba-Galea & Chiem, 2020). The existence of such transitions, well documented yet largely ignored for several decades, implies that the "open" state is likely meta-stable and hardly attainable with the current approaches employed in structural biology. These examples lead to the conclusion that channels adopt conformations beyond the simple opening and closing of their pores, and extreme care needs be exercised when interpreting or using structural data. Some help for interpreting the experimentally available structures may certainly come from atomistic simulation techniques that we are about to introduce." (cf. page 16/17).

Pag 22, third paragraph: As part of their second point, the authors should highlight that the paths for activation and deactivation are likely not the mutual reverse process. Instead, they follow unique step that can vary with activity. Examples like Shaker, Kv3.1, and hERG should be considered as instances of this topic.

We introduced the concept of hysteresis in the gating process with a new sentence in the second paragraph, with regard to the various predefined paths determined by the energy landscapes encountered: "These paths may even be different for activation and deactivation, with the elected pathway depending on the boundary conditions used (Villalba-Galea & Chiem, 2020). Shaker and hERG channels are pertinent examples of this occurrence." (cf. page 24/25).

Referee #3:

In their retrospective entitled "The 70-year search of voltage-dependent gating models in ion channels" Catacuzzeno and Franciolini review the progress on our understanding of the mechanism of voltage-gating since the groundbreaking work of Hodgkin and Huxley. Overall the manuscript is very well written and presents a clear historical narrative of the progress in this field.

Comments

The figure labeling and referencing needs to be made consistent throughout.

We have relabeled all the figures and their reference in the text.

There is a reference to Figure 1 on page 2 but there is no Figure 1. I believe this should be Figure 2.1.

Correct! We've fixed it.

The authors refer to Figure 3.1B in the text but never to Figure 3.1A.

The authors are inconsistent with their references to figures in the Boxes, for example, they refer to Figure B1 in Box 1 but there are no references to Figure B2 in Box 2 (same for Figures B5A and B5B in Box 5).

There is a reference to Figure 5.3 in Box 4 but no reference to Figure B4.

There is no reference to Figure 6.1 in the text.

The referencing to Figure 7.1A, Figure 7.1B and Figure 7.1C on page 20 are out of order (A, C then B). Either the order of the figure should be altered or the discussion in the text should be changed.

We have reviewed the entire Ms and made figure labeling and references consistent throughout.

The text on pages 12-13 needs to be better written to accommodate Box 4. It appears that the authors had written the text as a single section and then split some of it into Box 4 without altering the main text (this can be seen also from the Figure 5.2 reference error above). This results in the main text losing logical consistency without the reader having to first read Box 4 as it is presented in the pdf. This needs to be corrected. Either remove Box 4 and just keep it all as one continuous section (as it appears to have been originally written) or rewrite the main text so that it flows consistently without being dependent on the text in Box 4.

Correct diagnosis! We apologize for overlooking it in our final Ms reading. In any event, we have now fixed it by rewriting the main text to make it flows consistently without being dependent on the text in Box 4.

Page 2, paragraph 2, sentence 1: there is an extra comma that needs to be removed

Page 8, section 4.2, paragraph 1, sentence 1: "cysteine binding compounds, as menthanethiosulfonate (MTS)..." should be "cysteine binding compounds, such as menthanethiosulfonate (MTS)..."

BOX 2, sentence 1: "To accommodate its FRET results..." should be "To accommodate their FRET results..."

Page 20, paragraph 5, sentence 1: "As result..." should be "As a result..."

All the points above have been fixed. Thank you!

Page 22, the final sentence before "Outlook" section: "Brownian modeling, which simplified the systems studied by introducing sensible approximations as a tradeoff for creating the much needed macroscopic gating currents and their relevant parameters." is a sentence fragment, it is missing a verb, what did the Brownian modeling do? It is unclear but maybe this is what was meant ""Brownian modeling simplified the systems studied by introducing sensible approximations as a tradeoff for creating the much-needed macroscopic gating currents and their relevant parameters."

The reviewer is correct in noticing the lousy sentence, and kind in suggesting the way to fix it. So we did.

REQUIRED ITEMS:

-Please upload separate high quality figure files via the submission form.

Please notice that at this time we are not uploading high quality figure files because. Following indications from Mrs/Ms Sally Howells, managing Editor of JP, figures have not been finalized yet, as they will have to be first sent to professional illustrators for them to look stylish, once the review is (if) accepted.

Dear Dr Catacuzzeno,

Re: JP-TR-2022-282780R1 "Due 31 January: Topical Review: The 70-year search of the voltage sensor of ion channels" by Fabio Franciolini and Luigi Catacuzzeno

I am pleased to tell you that your Topical Review article has been accepted for publication in The Journal of Physiology, subject to any modifications to the text that may be required by the Journal Office to conform to House rules.

NEW POLICY: In order to improve the transparency of its peer review process The Journal of Physiology publishes online as supporting information the peer review history of all articles accepted for publication. Readers will have access to decision letters, including all Editors' comments and referee reports, for each version of the manuscript and any author responses to peer review comments. Referees can decide whether or not they wish to be named on the peer review history document.

The last Word version of the paper submitted will be used by the Production Editors to prepare your proof. When this is ready you will receive an email containing a link to Wiley's Online Proofing System. The proof should be checked and corrected as quickly as possible.

All queries at proof stage should be sent to tjp@wiley.com

The accepted version of the manuscript will be published online, prior to copy editing in the Accepted Articles section.

Are you on Twitter? Once your paper is online, why not share your achievement with your followers. Please tag The Journal (@jphysiol) in any tweets and we will share your accepted paper with our 22,000+ followers!

Yours sincerely,

Ian D. Forsythe
Deputy Editor-in-Chief
The Journal of Physiology
<https://jp.msubmit.net>
<http://jp.physoc.org>
The Physiological Society
Hodgkin Huxley House
30 Farringdon Lane
London, EC1R 3AW
UK
<http://www.physoc.org>
<http://journals.physoc.org>

*** IMPORTANT NOTICE ABOUT OPEN ACCESS ***

Information about Open Access policies can be found here <https://physoc.onlinelibrary.wiley.com/hub/access-policies>

To assist authors whose funding agencies mandate public access to published research findings sooner than 12 months after publication The Journal of Physiology allows authors to pay an open access (OA) fee to have their papers made freely available immediately on publication.

You will receive an email from Wiley with details on how to register or log-in to Wiley Authors Services where you will be able to place an OnlineOpen order.

You can check if you funder or institution has a Wiley Open Access Account here <https://authorservices.wiley.com/author-resources/Journal-Authors/licensing-and-open-access/open-access/author-compliance-tool.html>

Your article will be made Open Access upon publication, or as soon as payment is received.

If you wish to put your paper on an OA website such as PMC or UKPMC or your institutional repository within 12 months of publication you must pay the open access fee, which covers the cost of publication.

OnlineOpen articles are deposited in PubMed Central (PMC) and PMC mirror sites. Authors of OnlineOpen articles are permitted to post the final, published PDF of their article on a website, institutional repository, or other free public server, immediately on publication.

Note to NIH-funded authors: The Journal of Physiology is published on PMC 12 months after publication, NIH-funded authors DO NOT NEED to pay to publish and DO NOT NEED to post their accepted papers on PMC.

EDITOR COMMENTS

Senior Editor:

Some adjustments to language are needed; however, the copy editor can assist with this during production.

Minor changes suggested below can be made in the proofs.

Reviewing Editor:

Thank you for responding to all of the reviewer suggestions!

The Abstract is nice. One sentence is awkward:

"Further understanding on channels gating..." Possible alternatives:

"Further understanding of how channels gate..."

"Further understanding of the gating of channels ..."

"Further understanding of channel gating..."

REFEREE COMMENTS

Referee #1:

I am happy with the corrections. No further comments.

Referee #2:

The manuscript has been revised and greatly improved. However, some issues remain to be addressed. Those are:

Page 3, third paragraph: in fairness, Kenneth Cole and Howard Curtis (1939) proved the Julius Bernstein's hypothesis of membrane resistance breakdown. In doing so, Cole and Curtis described, before World War II, that a change in conductance across the membrane of the *Loligo* giant axon occurred during the generation of the axonal action potential. However, they did not identify the nature of the underlying conductances. Sodium was identified as essential by the work of Katz and Hodgkin just after WWII. With all these pieces of information, the leap forward in our understanding came with the work of Hodgkin and Huxley published in 1952 where they pinpointed the role of the Na⁺ and K⁺ conductances.

Page 5, third paragraph, 10th line: Shaker W434F is not an impermeant channel. Instead, it is a ultrafast inactivating channels that conducts Na⁺ in the absence of K⁺ (Starkus JG, Rayner MD, Heinemann SH. Anomalous conduction in the "non-conducting" Shaker K⁺channel mutant W434F. *Biophys J.* 1997;72:232a; Yang Y, Yan Y, Sigworth FJ. How does the W434F mutation block current in Shaker potassium channels?. *J Gen Physiol.* 1997;109(6):779-789. doi:10.1085/jgp.109.6.779)

Page 6, last paragraph: "(to the fourth power as the equation represents four parallel two-state models for the voltage sensor controlling the opening of a channel)" should read "(the equation is raised to the fourth power ...")

Page 7, second paragraph, last sentence: Citation is missing.

Page 8, second paragraph, first sentence: Papazian et al. citation is missing.

Page 11, second paragraph: The authors FRET instead of LRET when referring to several studies. FRET is a technique that measure either the amplitude or lifetime of fluorescence emission. LRET refers to a technique that measures the lifetime of phosphorescent emission from lanthanides. Although the basic principles are that of RET (resonance energy transfer), the implementations and readouts are very different. Hence, this paragraph must be revised.

Referee #3:

The authors have addressed all of my previous points satisfactorily. Find attached a pdf with some minor edits that I made by hand.

1st Confidential Review

06-Apr-2022

The Journal of Physiology

<https://jp.msubmit.net>

JP-TR-2022-282780

Title: Due 31 January: Topical Review: The 70-year search of the voltage sensor of ion channels

Authors: Fabio Franciolini
Luigi Catacuzzeno

Author Conflict: No competing interests declared

Author Contribution: Fabio Franciolini: Conception or design of the work; Acquisition or analysis or interpretation of data for the work; Drafting the work or revising it critically for important intellectual content; Final approval of the version to be published; Agreement to be accountable for all aspects of the work Luigi Catacuzzeno: Acquisition or analysis or interpretation of data for the work; Drafting the work or revising it critically for important intellectual content; Final approval of the version to be published; Agreement to be accountable for all aspects of the work

Running Title: The 70-year search of the voltage sensor of ion channels

Dual Publication: No

Disclaimer: This is a confidential document.

Funding: NO ~~F~~OUNDING: Luigi Catacuzzeno, N/A

FUNDING

The 70-year search of voltage-dependent gating models in ion channels

Luigi Catacuzzeno and Fabio Franciolini

Department of Chemistry, Biology and Biotechnology, University of Perugia (Italy)

A retrospective of the scientific journey initiated in 1952 with the 'gating charged particles' postulated by Hodgkin and Huxley, later visualized as gating currents. Subsequently the S4 segment was identified as the molecular counterpart of these particles, visualized in the crystallographic structures of voltage-dependent channels and investigated by molecular dynamics simulations, in the quest for gating models of voltage-dependent ion channels.

1. Introduction

Our journey begins 70 years ago, in 1952, with the publication of a series of papers in the Journal of Physiology by Alan Hodgkin and Andrew Huxley, that represented a milestone in our understanding of neuronal excitability (Hodgkin & Huxley, 1952*a*, 1952*b*, 1952*c*, 1952*d*). The conceptual and experimental work that underpinned these papers began in the Summer of 1947, at the Laboratory of the Marine Biological Association, Plymouth, with Hodgkin and Huxley using the voltage clamp technique to record the voltage-dependent Na and K currents from the giant axon of the squid, *Loligo*. During the following few years, at the Physiological Laboratory in Cambridge, they carried out the analysis of these transmembrane currents, their voltage and time dependence, and conducted a mathematical model of the axonal action potential that described accurately its time course, threshold, propagation, and refractoriness. This monumental work remains one of the most extraordinary conceptual achievements in biophysics and biology. For their work Hodgkin and Huxley received the Nobel Prize in Physiology or Medicine in 1963.

How and where these currents crossed the membrane at that time remained unresolved. In their papers Hodgkin and Huxley referred only to ion currents and conductances, never to ion channels. Indeed, the concept of an ion channel was for the future, and a structural correlate nonexistent. Some would think of holes in the membrane, others of cracks forming upon voltage changes as likely pathways for the ion fluxes. Functional studies in the 1960s, mainly contributed by Bertil Hille, Clay Armstrong, Bernard Katz, Hans Meves and several others, laid the foundation for the notion of Na⁺ and K⁺ ions crossing the membrane through aqueous pores provided by specific proteic structures: the ion channels (Meves & Chandler, 1965; Katz & Miledi, 1966; Moore, 1967; Hille, 1968; Armstrong, 1971).

2. Hodgkin, Huxley and the ‘gating charged particles’

In their experiments Hodgkin and Huxley found that the total membrane current activated by depolarization in the giant axon of the squid could be separated into two components, the Na and the K current, I_{Na} and I_K . They studied these individually for their dependence on membrane potential and time and by introducing the parameter conductance – g_{Na} and g_K for Na^+ and K^+ ions – linked the currents to their driving force ($V - E_x$), with simple ohmic relation: $I_K = g_K (V - E_K)$ and $I_{Na} = g_{Na} (V - E_{Na})$. In this way g_{Na} and g_K could be readily determined as they both coincided with the time course of I_{Na} and I_K in response to a voltage step, under the assumption that during the few ms of the imposed voltage pulses E_x would not change.

Looking first at the time dependence of g_K , which is easier to describe, on applying a depolarizing step it rises slowly following a sigmoidal time course, until it reaches a steady level. On repolarization g_K falls instead exponentially towards zero, and with much faster time course than the rising phase (Figure 1). A sigmoidal rise of g_K and an exponential fall was suggestive, as Hodgkin and Huxley pointed out, of the K conductance being controlled by a number of independent ‘gating particles’ (in Hodgkin and Huxley terminology), in a way that all of them had to be in the ‘permissive’ position for the membrane to pass K^+ ions (the sigmoidal rise), but only one of them switching back to the ‘non-permissive’ position was sufficient to make the membrane impermeant to K^+ ions (the exponential fall). As we indicated, at the time Hodgkin and Huxley proposed their model there was no notion of ion channels as we now know them. Nonetheless, on behalf of novices to the field, from here on we will refer and describe their model in terms of ion channels, as we now know them.

Figure 2.1 - Hodgkin and Huxley and the gating charged particles. The figure, from one of their 1952 papers, shows the K conductance time course of a squid giant axon membrane upon depolarization from and then repolarization to the resting potential. [From Hodgkin & Huxley, 1952d]. *Inset:* Sketch illustrating the relocation of the postulated gating charged particles across the membrane upon depolarization. [From Catacuzzeno et al., 2019].

From fitting g_K at varying voltages, Hodgkin and Huxley found that a good fit was obtained when the number of gating particles, n , was made equal to four, giving a probability that all the four gating particles are in the permissive position equal to n^4 . This notion can be analytically represented by factorizing this parameter in the previous ohmic relation, to obtain

$$I_K = g_{K,max} \cdot n^4 \cdot (V - E_K)$$

Because the opening of K channels depended on membrane potential, the postulated gating particles were assumed to be electrical charges or dipoles that relocate within the membrane in a voltage dependent manner. Supposing further that each particle moves, in response to voltage,

from its non-permissive (resting) to permissive (activated) position and back with first order kinetics, the distribution of the charged particles as described by the probability n will follow an S-shaped curve (n^4), as does the delayed rise of g_K on depolarization, whereas it falls exponentially to zero, mirroring the decrease of g_K on repolarization.

A corresponding conceptual approach was used for the Na current, with the contingency that the Na current turned off spontaneously for the inactivation process. Hodgkin and Huxley accounted for this by introducing the inactivation particle, h , that would switch the Na conductance off, with maintained depolarizations. With the Na current, they obtained a good fit when the number of gating particles (m) was made equal to three, giving a probability that all the three particles are in the permissive position equal to m^3 .

With their studies Hodgkin and Huxley accurately described the ionic currents underlying the action potential, and predicted its major properties including time course, propagation and refractoriness. They also proposed the key elements of the gating mechanism of the 'future' ion channels when these were only a concept at best: namely, the *gating charged particles* that are required to move across the membrane electric field to switch on and off g_{Na} and g_K , i.e., the opening and closing of ion channels. This was a signpost to the biophysicists of the time the direction to go: towards the gating currents that these gating charged particles would generate with their movement.

3. The gating currents

Because of their small size and fast kinetics, and in addition being combined with fast ion currents (such as the Na current) and the capacitive currents needed to charge the membrane to new potentials, the first recordings of the postulated gating currents had to wait more than 20 years since Hodgkin and Huxley had predicted them. They were first recorded from skeletal muscle (Schneider & Chandler, 1973) and soon after from the squid giant axon (Armstrong & Bezanilla, 1973; Keynes & Rojas, 1974), and were arguably linked to the activation of the Na channels. In the mid 1980s the cloning of Na and K channels and their expression at high density in *Xenopus* oocytes allowed gating currents to be recorded also from K channels, and mutations to be inserted into the channel proteins, which started the investigation of the gating currents underpinning: the voltage sensor. The observation that mutation W434F in the pore region of Shaker rendered the channel fully impermeant to K^+ ions, while preserving virtually unchanged the gating current and its main features (Perozo *et al.*, 1993). This represented a further important improvement in the study of the gating currents, and became the standard model for investigating the gating currents of K channels and the structure-function relationship of its voltage sensor.

Figure 3.1 - Gating currents. **A)** *Top:* Armstrong and Bezanilla's first gating current recording from the squid giant axon elicited in response to a 2.5-ms step to 20 mV from -120 mV, in zero external Na^+ , and TTX added. *Bottom:* Simultaneous recording of Na current with external Na^+ concentration reduced to 10 mM. [From Armstrong and Bezanilla, 1974]. **B)** Kinetic scheme for the four voltage sensors derived from the analysis of the gating current of Shaker channels. [From Catacuzzeno et al., 2020].

3.1 - Gating currents neither rise instantaneously nor decay monoexponentially

As soon as the first gating currents were recorded, their characteristics demonstrated that their rise was not instantaneous, as would have been expected from the Hodgkin and Huxley's two-state model of the gating charged particles, but rather showed a clear rising phase. Likewise, the decay phase could not be described by a single exponential function, as anticipated from the same model, but instead comprised several components (Armstrong & Bezanilla, 1977; White & Bezanilla, 1985). Both the sigmoidal rising phase and the multi-component decay of the gating currents were thought to depend on more complex kinetics of the gating charge translocation than expected from the two-state model initially proposed. Several states in the activation pathway of the gating particles were later established by modeling experimental gating currents with discrete state rate (Markov) models (Schoppa *et al.*, 1992; Tytgat & Hess, 1992; Bezanilla *et al.*, 1994; McCormack *et al.*, 1994; Zagotta *et al.*, 1994).

State rate modeling of Shaker gating currents revealed several properties of the gating process. First, the four voltage sensors of the channel – the tetrameric architecture of voltage-gated channels had been now established (MacKinnon, 1991) – would move for quite a long portion of their journey mostly independently, as shown in the kinetic scheme of Figure 3.1B. It was found that at least three states were needed to reproduce the rising phase, with an initial transition that translocates less charge or has a smaller forward rate constant than the following transitions (Taylor & Bezanilla, 1983). On the contrary there was instead a high level of cooperativity between the four voltage sensors in the final step that would open the channel. Second, the voltage sensors would move in several steps and distinctive transition rates to account for the initial rising phase and subsequent multiexponential decay.

Also at variance with the original views was the decay time course of the OFF gating current which, according to Hodgkin and Huxley's model, ought to be three times faster than the ion current decay time (this is because only one particle was required to switch back to the resting position to close a Na channel). Experimental data showed instead that the gating current and the Na current decay with comparable time courses (cf. Armstrong and Bezanilla, Phil Trans 1975). These results indicate that the return of the three independent gating particles to their resting position, associated with the channel closure, follows a more complex kinetics.

Taken together, these data show that gating currents are more complex than the two-state gating charged particles initially proposed (when the gating currents had yet to be recorded!). The original model clearly required revision in several critical aspects to account for the new features of the gating currents that were being discovered (Armstrong & Gilly, 1979).

3.2 - Counting the gating charges associated to a single channel opening

A major parameter that quantifies the voltage sensitivity of an ion channel is the amount of gating charge that needs to move across the membrane to open the channel. This quantity can be

estimated from the equilibrium open probability, p_o , of a voltage-gated channel, that takes the form of a sigmoidal function of membrane potential (to the fourth power for a channel displaying four gating particles or voltage sensors): $p_o = \{1/(1 + \exp[-z_g \cdot e \cdot (V - V_{1/2})/k_B T])\}^4$. Here the quantity $-z_g \cdot e \cdot (V - V_{1/2})$ represents the electrical energy increase associated with the transition of a gating charge from the non-permissive to permissive position, and z_g is the gating charge translocated by a single particle that can be measured from the slope of the $p_o/(V)$ curve.

These concepts were originally used by (Hodgkin & Huxley, 1952d) to gain information on the gating charge associated with the increases in Na and K conductance, which provided values of 6 and 4.5 elementary charges, e_0 , that need to move across the whole membrane electric field to open a Na and a K channel, respectively. They stressed however that they ought to be taken as minimum values, because in principle that approach is applicable only to a gating charge that translocates between the non-permissive and permissive positions in a single step. Subsequently several studies have demonstrated that the gating kinetics of most voltage-dependent channels, including Na and K channels, involve more than just two states, each associated with the translocation of varying amounts of gating charge.

In 1978 (Almers, 1978) developed a method – the *Limiting slope method* – to assesses the gating charge that is necessary to move to open a voltage-dependent channel also when the movement of the charge occurs in small packets over a sequence of several closed states, thus allowing the applicability of the method to channels with multi-state kinetics. The method requires assessment of the $p_o/(V)$ relationship at very negative voltages, where p_o can truly be considered to approach zero, and the $p_o/(V)$ relation be linear (the *Limiting slope*), to avoid underestimates of the effective gating charge, z_g , a condition not easily met. Using this method to assess the total charge movement per channel, several groups have reported consistent values of z_g for Na and K channels, which ranged between 12 and 16 e_0 (Almers & Armstrong, 1980; Bezanilla & Stefani, 1994; Zagotta *et al.*, 1994; Hirschberg *et al.*, 1995).

Estimates of the gating charge associated with the opening of a single channel can also be obtained from the ratio between the total gating charge, Q , moving upon maximal activation (estimated as the time integral of the gating current), and the total number of contributing channels (N , estimated with either noise analysis or toxin binding). This method requires independent assessments of Q and N made on the same preparation, on which the ratio between the two quantities, (Q/N) , is to be calculated. Estimates of the gating charge per channel, carried out with this method on gating currents from K channels expressed on *Xenopus* oocytes, were very consistent falling between 12 and 14 e_0 , with no apparent bias between the two methods to estimate the number of channels (noise analysis or toxin binding) (Schoppa *et al.*, 1992; Zagotta *et al.*, 1994; Aggarwal & MacKinnon, 1996; Noceti *et al.*, 1996; Seoh *et al.*, 1996; Islas & Sigworth, 1999).

4. The primary structure of the Na and K channels is discovered

In 1984 the voltage-gated Na channel from electroplax membranes of *Electrophorus electricus* was cloned and its primary structure determined (Noda *et al.*, 1984). The Na channel protein was made of a single polypeptide chain of more than 1,800 residues that folds to form four homologous domains (I–IV), each displaying six transmembrane α -helical segments (S1 to S6) (Figure 4.1A). Of

particular interest, one segment in each domain – segment S4 – had 4 to 7 positive charges invariably separated by two non-charged residues. This peculiar and consistent concentration of positively charged residues in the four S4 segments immediately suggested that it might be serving as the voltage sensor in Na channels (Noda *et al.*, 1984; Greenblatt *et al.*, 1985). The elucidation of the primary structure of the Na channel and the discovery of the highly charged S4 segment led to the development of the *Sliding helix* model to explain the gating mechanism of voltage-gated channels (Box 1).

Shortly afterwards the Shaker K channel from *Drosophila* and Ca channels from various muscle types were cloned (Tanabe *et al.*, 1987; Tempel *et al.*, 1987; Kamb *et al.*, 1988; Pongs *et al.*, 1988; Mikami *et al.*, 1989; Koch *et al.*, 1990), and their overall architecture was found to mirror that of Na channel's, with four modules (independent protein subunits for Shaker and four domains in a long amino acid stretch for Ca channels), each containing six transmembrane α -helical segments (S1–S6), of which S4, like the Na channel's, contained an excess of positive charges systematically separated by two non-charged residues (see Figure 4.1B for Shaker), further corroborating the notion that this was the voltage sensor of voltage-gated ion channels.

Figure 4.1 - Primary structures of the voltage-gated human Nav1.4 and Shaker K channels. Typical six-transmembrane α -helical segments (S1–S6) architecture of each domain (of Nav1.4) or subunit (of Shaker) of voltage-gated channels, showing the segment S4 to contain an excess of positive charges invariably separated by two non-charged residues (see segments S4's sequences in insets). Basic residues on S4 are shown in blue, and acidic residues on S2 and S3 in red. [Modified from Bezanilla, 2008].

BOX 1 – The *Sliding helix* model

In 1986, Catterall and separately Guy and Seetharamulu converged on the *Sliding helix* (or *Helical screw*) model for the voltage sensor movements during channel gating (Guy & Seetharamulu, 1986; Catterall, 1986). The model depicts the the S4 segment, with its positive charges placed along a spiral strip, to be pulled inwards at negative (resting) potential by electrostatic forces, and stabilization of the charged voltage sensor in the hydrophobic environment to be reached by the interaction of the positive charges on S4 with negatively charged residues on nearby helices (Figure B1). Release of these inward-directed forces that occur with depolarization makes the S4 segments move outwards, and the positive charges interact in succession with countercharges on segments S1-S3. Maximal displacement of S4 segments was estimated in 13.5 Å, with a rotation of

~180°. The interaction of positive and negative charges, also assumed by the model, gave solution to the thermodynamic dilemma of inserting highly charged structures – the S4 segment – into the extremely hydrophobic environment of the plasma membrane, and the sequential formation of ion pair explained the energetics of the S4 outward movement during activation.

Figure B1 - The Sliding helix model of gating. Sketch of the Sliding helix model suggesting that the positive charges, arranged in a spiral shape, pull the S4 segment inwards in the resting state (at negative potential inside). Following activation the inward-directed forces on the S4 segment are released and the segment is pushed outwards, allowing the positive charges to pair in succession with the negatively charged residues on the vestibule walls (shown as red dot). [From Catacuzzeno et al., 2020].

4.1 – Testing the S4 segment as the voltage sensor of voltage-gated channels

Conti, Stühmer and coworkers set out to test the primary prediction of the Sliding helix model for the voltage-dependent Na channel. They reasoned that if the positively charged residues on S4 really are the gating charges of the voltage sensor, their neutralization should reduce the channel's voltage sensitivity. Using site-directed mutagenesis, they assessed the voltage sensitivity of Na channels expressed in *Xenopus* oocytes when one or more positive charges on S4 of domain I were replaced by neutral or negative residues (Stühmer *et al.*, 1989). The reduction by varying amounts of the overall net positive charge induced a corresponding decrease of the apparent gating charge z_g and a rightward shift of the voltage dependence curve. These data supported the concept that S4 was the voltage sensor of the Na channel, although they did not represent a conclusive demonstration. More compelling experimental data came with the cloning of the Shaker K channel (Papazian *et al.*, 1987) which was more extensively studied because of its much smaller size (~1/4 that of the Na channel), and thus more favorable for genetic manipulations.

The effects of neutralizing the various positive charge on the Shaker S4 segment were found to result in a seemingly concurrent decrease of the K current's voltage sensitivity, first reported by Logothetis *et al.* (1993). Some inconsistencies however emerged, as it was found that mutations of differently located positive charges could have very distinct effects, indicating that the channel's voltage dependence could not be fully explained by electrostatic considerations only. It needs be recalled however that taking the channel current's voltage sensitivity as a readout of what is happening in another channel's domain – the VSD – may well lead to misevaluation.

Two groups addressed this issue directly by assessing the gating charge per channel after charge neutralization on Shaker's S4 segment (Aggarwal & MacKinnon, 1996; Seoh *et al.*, 1996). They both found a significantly lower gating charge translocation upon channel activation when

they neutralized the first four S4 residues R1-R4¹, as compared to K5 or K7, which only showed a little difference with the WT². Overall the data from these two studies indicate that the first four basic residues R1-R4 are those principally involved in generating the gating current, and thus important in gating (Aggarwal & MacKinnon, 1996; Seoh *et al.*, 1996; Bezanilla, 2002).

A second expectation of the *Sliding helix* model was that the gating charges on S4, while moving upwards through discrete states upon channel activation, would generate miniature currents. The size of these 'shot' currents, as they came to be known, was below the threshold of resolution of electrophysiological instrumentation. They would however create fluctuations on the gating current recorded from a population of channels, and the analysis of these fluctuations would provide information on their features. Conti and Stühmer expressed rat brain Na channels in *Xenopus* oocytes at a high density, and analyzed the fluctuations (noise) of the gating currents. By applying this approach they found that: *i*) the autocorrelation of the fluctuations was consistent with a shot-like type of the gating charged particles movement (thus excluding that the charged particles move in a diffusional regime); *ii*) the movement of the gating charge when a single Nav channel is activated displays a major step carrying an equivalent of $\sim 2.3e_0$. Several years later Bezanilla's group, using a similar approach, found similar values for the Shaker K channels ($2.4e_0$; (Sigg *et al.*, 1994)). These authors suggested that this large quantal charge was possibly associated with a late step in the S4 translocation upon activation³.

4.2 – The *Sliding helix* model predicts that S4 moves outward upon activation

Voltage-dependent movement of S4 was initially tested on Nav channels by using the substituted cysteine accessibility method that involves replacing specific residues on S4 with cysteine, and probing if cysteine-binding compounds, as methanethiosulfonate (MTS), can reach these cysteines from either side of the membrane, as a means of defining the position of S4. MTS binding to substituted cysteines could be visualized as modifications of voltage-dependent gating. In the first group of experiments, cysteine was replaced for the first (most outward) arginine residue R1 (R1448) in the S4 of domain IV of skeletal muscle Nav channels (Yang & Horn, 1995). External application of MTS did not affect the Na current when the membrane was held at negative (resting) potentials. By contrast, it drastically modified the Na current kinetics (the inactivation rate) when the membrane was depolarized, as would be expected for a voltage sensor moving outward with depolarization (Yang & Horn, 1995). In further studies cysteine replaced arginine at positions R2 and R3 of the same Nav channel S4 segment. They found that cysteines at these positions could be accessed by MTS reagents applied intracellularly when the cell was

¹ In fact, (Seoh *et al.*, 1996) were not able to assess the contribution of R1 to channel gating.

² Neutralization of residue R6 (R377) resulted with no viable channels in the plasmamembrane. However, a later study with histidine scanning mutagenesis showed that R377 (and also K374) do not participate in gating (i.e., they do not enter the transmembrane electric field) (Starace & Bezanilla, 2001).

³ Few years later, on reexamining Conti and Stühmer's data on Nav channels in the attempt to generalize their conclusions to other voltage-gated channels, (Crouzy & Sigworth, 1993) suggested that the $2.3e_0$ obtained by Conti and Stühmer, and likewise the $2.4e_0$ later obtained by (Sigg *et al.*, 1994), might not reflect the true size of the actual charge crossing the pore. Because of the limited filter bandwidth used in Conti and Stühmer's experiments (8 kHz), a rapid passage of consecutive unitary (gating) charges through the pore would not appear as single passages, but become indistinguishable from a single large charge movement. In other words, an experimental artifact due to limited filter bandwidth.

hyperpolarized, while they were reached from the outside following depolarization (Yang *et al.*, 1996).

A similar approach was used in Isacoff's lab to probe the movement of S4 segment of Shaker channels (Larsson *et al.*, 1996; Baker *et al.*, 1998). They replaced several residues, including three gating charges (R362, R365, R368), over a stretch of 18 amino acids (from A359 to S376) of the Shaker S4 region, and tested their intracellular and extracellular thiol reagent accessibility as function of voltage. Large distal portions of S4, with the exclusion of only a short central part, were found to be accessible to the thiol reagent, indicating that S4 can slide into the internal and external vestibules, in a voltage-dependent manner. Taken together these results indicate that at hyperpolarized (resting) potentials S4 protrudes into the intracellular vestibule with all but the first gating charge (R1), while it moves outwards, upon depolarization, to expose the three outermost gating charges, R1-R3, to the external vestibule. These studies gave indirect estimates of the S4 translocation upon activation in the range 8 to 15 Å,

To rigorously assess the S4 movement during gating, Bezanilla and coworkers used the fluorescence resonance energy transfer (FRET) technique that can accurately measure the distance between two light-sensitive molecules even when they move relative to each other by only few Å (Cha *et al.*, 1999; see also Glauner *et al.*, 1999). They initially attached the chromophores at similar positions on the S4 segments of different subunits of Shaker. With this arrangement, they recorded FRET signals during voltage gating consistent with a S4 rotation of 180° (Cha *et al.*, 1999; Glauner *et al.*, 1999). To estimate the translocation of S4 with respect to the surface normal of the membrane, they fixed one chromophore to a toxin bound to the pore mouth of the channel and the other to the upper section of the S4 segment. These experiments showed a translocation of S4 less than 2 Å (Chanda *et al.*, 2005; Posson *et al.*, 2005), clearly in contrast with both the *Sliding helix* model and many experimental data. This clearly called for some new mechanism of gating that could explain how such a small S4 translation could account for the 12-14 e_0 per channel translocated with full channel activation (Box 2). Additional FRET experiments, probing a higher number of residues, indicated that the translocation of the voltage sensor was in fact significantly higher and closer to 10 Å (Posson & Selvin, 2008), a result more consistent with a large array of data and the *Sliding helix* model.

their

BOX 2 – The *Transporter* model

To accommodate its FRET results, Bezanilla and coworkers proposed the *Transporter (rotational)* model that pictures the aqueous crevices from both sides of the membrane penetrating deep into the VSD, in a way to shape the electric field around the S4 segment more parallel to the plane of the membrane. According to the model the first four gating charges on segment S4 (R1-R4, those important for channel gating) are exposed to the narrow aqueous intracellular crevice under hyperpolarized conditions, but to the extracellular crevice upon depolarization, as result of a 180° rotation of segment S4, with a very minor translation relative to the plane of the membrane. Assuming a focused electric field across the S4 segment, the model was able to account for the 12-

13

14 e_0 moved across the electric field upon VSD activation without any significant translational movement of the helix (Cha & Bezanilla, 1998; Cha *et al.*, 1999; Bezanilla, 2002).

Figure B2 - The *Transporter* model. The first four charges of the S4 segment, R1-R4, are exposed, under hyperpolarized conditions (left), towards the narrow aqueous intracellular crevice, but towards the extracellular crevice under depolarized conditions (right). According to the model, the depolarization translocates the four gating charges from the internal to the external crevice, with a ca. 180° rotation of S4, without any significant translation normal to the plane of the membrane. The model requires a highly focused electric field between the internal and external crevices. [Homemade].

5. X-ray crystallographic pictures of K and Na channels

By the turn of this century the basic principles of voltage-dependent gating had been consolidated: the S4 segment had been demonstrated to be the voltage sensor; the positively charged amino acids critical for channel gating (R1-R4), identified; the number of gating charges (12-14 e_0) that move across the membrane electric field (by 10-15 Å) during channel gating established; the general movement of the voltage sensor through the gating pore and the zipper-like interaction with VSD walls determined. The next step was to extend these findings with high-resolution structural evidence that in early 2000 began to appear as X-ray crystallographic three-dimensional molecular structures of several voltage-gated channels. The research activity of voltage sensing in ion channels in this period was rewarded by unprecedented advances.

5.1 – Crystallographic structure of the KvAP channel

The first X-ray crystallographic structures of a voltage-gated ion channel was reported by MacKinnon's lab for the bacterial K channel KvAP, from the thermophilic *Aeropyrum pernix* (Jiang *et al.*, 2003b; Ruta *et al.*, 2003). It confirmed the tetrameric architecture of the voltage-gated K channels derived from functional studies, with the S5 and S6 segments of the four subunits forming the permeation pore, and the S1–S4 segments making up the VSD. The four VSDs, which displayed both sequence and properties similar to their eukaryotic counterparts, appeared located at the periphery, loosely joined to the pore domain by S4–S5 linkers. This structure was however unexpected in several respects and inspired a voltage-gating model distinct from both the *Sliding Helix* and *Transporter* models (Box 3).

BOX 3 – The *Paddle* model

In the KvAP the segment S4 was found tightly bound to the S3b (the second half of segment S3, which is bent in the middle) to form the S3b-S4 helical hairpin, and this paired structure was found dislocated deep inwards near the protein-lipid interface and nearly parallel to the surface of the lipid bilayer (Figure Box-A). These observations, combined with structural data of the isolated voltage-sensing module showing interactions of the innermost gating charges with negative groups in the lower portion of S2 (Jiang *et al.*, 2003b, 2003a), suggested a new gating mechanism – the *Paddle* model of activation – in which the S4 segment, during activation, moves considerably through the phospholipid matrix to translocate the estimated 12-14e₀ across the membrane electric field (Figure B3).

Figure B3 – The *Paddle* model. Side views of the KvAP crystal structure showing the spatial relation between the S4 and S3b, together forming the ‘paddle’, and with respect to the membrane in the hyperpolarized/resting (A) and depolarized/activated (B) conditions. [Modified from Swartz, Nat Rev Neurosci 2004].

Later structures of Kv channels showed an architecture different from the KvAP, which made scientists think that the KvAP channel had been significantly altered by the crystallization procedure, such that it no longer represented its native state (Lee *et al.*, 2005). The most likely reasons underlying the distorted structure of KvAP was reasoned to be its being crystallized in the absence of lipids, and with an antibody bound to it (Long *et al.*, 2005a, 2005b, 2007; Lee *et al.*, 2005).

5.2 – The crystal structures of Kv1.2 and Kv1.2/2.1 chimera

The crystal structure of KvAP was shortly followed by the determination of the structure of the voltage-gated K channels Kv1.2 (Long *et al.*, 2005a, 2005b) and Kv1.2/Kv2.1 chimera, comprised of the Kv1.2 channel and the voltage sensor from Kv2.1 channel (Long *et al.*, 2007). These high-resolution structures of the Kv channels presented the same architecture as KvAP, highlighting the key elements for channel gating and reinforcing the findings of previous studies. Namely, the S1–S3 segments form an hourglass-shaped structure with a central gating pore, a short and narrow hydrophobic constriction (or hydrophobic seal; HCS) essentially impermeant to water and ions that separates the internal and external aqueous vestibules and serves to focus virtually the whole transmembrane electric field to a range of only few angstroms (Yang *et al.*, 1996; Starace & Bezanilla, 2004; Tombola *et al.*, 2005; Chanda *et al.*, 2005).

In the Kv1.2/Kv2.1 chimera, whose VSD structure is shown in Figure 5.2, the high-resistance hydrophobic seal has a phenylalanine at 233 (F233) as a key residue (F224 in Kv1.2). On both sides of the gating pore the Kv1.2/Kv2.1 chimera VSD displays two negative charge clusters, one on the

extracellular side (ENC formed by E183 on S1 and E226 on S3) and one on the intracellular side (INC formed by D259 on S3 and E236 on S2) (Figure 5.2). These charge clusters were much expected, as already postulated in the *Sliding helix* model to serve as transitory ion partners for the movement of the charged voltage sensor S4. The crystal structure also shows that R4 interacts with the ENC (namely with E226), while the negative residues D259 and E236 of the INC trap the gating charge K5. This arrangement indicates that crystallization captured the VSD of Kv1.2/2.1 chimera in the activated state, i.e., strongly pushed outward, with the four gating charges R1-R4 on the external side of the hydrophobic seal, as expected given that the crystallization procedure cancels the potential across the membrane.

Figure 5.2 - Ribbon representation of the four α -helical structures of one VSD of Kv1.2/2.1 chimera in the activated state. A) Stick representation of the positive gating charges on the S4 segment (R0-R4 and K5, in yellow), the negative residues E183 on S1 and E226 on S3 (red), forming the ENC, and the more intracellular negative cluster, forming the INC, made by the residues D259 and E236. The central part of the VSD, forming the gating pore, or HCS, is essentially contributed by a phenylalanine residue (F233, green). Because this structure is common to several Kv channels, in parenthesis we give the corresponding residue numbers (only numbers, as the residue types are the same as in the VSD of Kv1.2/2.1 chimera) found in Shaker channel, where much investigation has been carried out. [From Catacuzzeno et al., 2020]. **B)** Sequence alignment of S2 and S3 from various channels. Highlighted are the conserved residues investigated here.

5.3 – The *Gating charge transfer center (GCTC)* model

Reflecting on the crystal structure of the Kv1.2/Kv2.1 chimera, with the first four arginines of S4 (R1-R4) on the external side of the hydrophobic seal and the next gating charge, lysine (K5), trapped in a putative binding site made by the evolutionary conserved phenylalanine F233 and the negative charges of glutamate and aspartate on S2 and S3 (E236, D259) (Figure 5.2), MacKinnon and coworkers considered that this specific grouping of charged amino acids would serve a major role in gating. To address this point they mutated the phenylalanine of the Shaker's VSD with several substitutes, and measured the resulting K currents from *Xenopus* oocytes, the expression system used for the mutated channels (Tao et al., 2010).

BOX 4 – The *Gating charge transfer center (GCTC)* model

As result of their investigation MacKinnon and coworkers proposed the *Gating charge transfer center (GCTC)* model, according to which, before crossing the gating pore the gating charges on the S4 segment transiently bind to the GCTC, made of the aromatic phenylalanine (F290) and the two negative residues E293 and D316 (Figure 5.3; (Tao et al., 2010)). By solvating and stabilizing in

succession the gating charges, the GCTC would facilitate the movement of S4 during gating, that is, would transiently stabilize the S4 gating charges before traversing the gating pore. According to this view, the first gating charge, arginine R1 ought to be found in the GCTC in the fully resting state (at very negative voltage), whereas when fully activated (at very positive voltage) lysine K5 should sit in the GCTC.

Figure B4 – The Gating charge transfer center (GCTC) model. A) Scheme showing the gating charge transfer center hypothesis, with each state characterized by a different gating charge present in the Gating Charge Transfer Center formed by the aromatic phenylalanine at 290 and the negative residues E293 and D316. For clarity the negative residue D316 on S3 has been omitted. The three state shown, C4, C3 and C2, are from Henrion *et al.*, 2012, with C4 representing the ‘deep closed state’ at most negative potentials, with the first gating charge, R1, in the GCTC.

The GCTC role in channel gating was challenged by the Ahern lab based on the observation that a neutral synthetic substitute at either E293 or D316 of Shaker did not appreciably modify the conductance-voltage curve (Pless *et al.*, 2011). The two residues E293 or D316 were in addition suggested to be located in the water-filled intracellular vestibule, thus outside the transmembrane electric field, greatly weakening their proposed role. On this basis, they suggested that some role in assisting channel gating could only be played by the highly conserved phenylalanine, by establishing transient gating charge- π interactions during S4 movement (Pless *et al.*, 2011).

The GCTC notion was also questioned on the grounds that it was essentially based on the effects of F290 mutants on the K currents, although ion currents are known to not exactly mirror what is going on at the voltage sensors. Lacroix and Bezanilla verified the effects of F290 mutants on the gating currents of Shaker, that is, the direct object of the mutants action. Out of the 13 mutations tested, only one – F290W – exhibited a meaningful effect on the gating charges (Lacroix & Bezanilla, 2011).

These experiments rolled back the enthusiasm for the GCTC view. However when the first crystal structure of a bacterial Na channel was disclosed, a structure absolutely similar to the GCTC of Shaker was also found in this channel (Payandeh *et al.*, 2011). Specific tests to address its function suggested however that the negative residues in the Nav channel GCTC seemed important for VSD structuring and channel trafficking to the membrane, leaving its role open to question (see below).

5.4 - The first crystallographic structure of a voltage gated Na channel is revealed

In 2011 Catterall's lab reported the first high resolution crystallographic structure of a voltage-gated Na channel (Payandeh *et al.*, 2011). The channel – the bacterial NavAb from *Arcobacter butzleri* – was made of four identical subunits, similar to that of voltage-gated K channels, and each subunit mirrored a single domain of mammalian Nav channels, with the six transmembrane segments making the VSD (S1-S4) and the pore domain (S5 and S6). The S1–S3 segments of the VSD were found to form an hourglass-shaped structure, along the lines of the Kv channels' VSD, with a short and narrow hydrophobic constriction (hydrophobic seal or plug) in its central part, essentially impermeant to water and ions, that separates the intracellular and the extracellular aqueous vestibules and focuses virtually the whole transmembrane electric field (Yang *et al.*, 1996). This hydrophobic seal again includes a phenylalanine as its most characterizing residue, showing its high evolutionary conservation. On both sides of the hydrophobic seal the NavAb VSD displays two negative charge clusters that earlier disulfide cross-linking studies using paired substituted-cysteine residues had already shown on a Nav channel homologue of NavAb, the NaChBac from *Bacillus halodurans*.

The highly conserved internal cluster made of the aromatic phenylalanine F56 and the two close negatively charged D80 and E59, have been suggested to be implicated, as in the Shaker channel, in the stabilization of the S4 charges, while the S4 segment steps up through the gating pore during activation. The proposed function of these structures of transiently binding the gating charges on S4 before they cross the gating pore during activation, has been however questioned on the grounds that in the Nav1.4 channel these residues have been reported to serve, as indicated above, for its proper folding and membrane trafficking. An alternative function proposed for these structures is to shape the potential profile across the membrane, and as result establish the kinetics of the S4 motion. In particular, the wider hydrophobic seal of the VSD of domain IV of Nav1.4 channel compared to the other three VSD, as reported by Gosselin-Badaroudine *et al.*, (2012), has been taken to explain its slower activation (Chanda & Bezanilla, 2002).

In concluding this section we are mindful that structural data taken by themselves can and often are misleading when translated directly into functional terms. Channels structures obtained by crystallization may only poorly reflect the actual structure of the native channels in the cell membrane (due to preparatory procedures for protein crystallization that rarely if ever preserve the natural biological environment). Thus extreme care should be exercised when interpreting or using structural data.

6. *Ab initio* and MD simulations to picture the closed state and the voltage sensor movement

As virtually all crystal structures reported so far have portrayed voltage-gated channels in the open/inactivated state, much effort was rapidly focused to understand the structure of the resting VSD. This knowledge would also provide clues on the conformation transitions associated with the gating process. Indeed, models of VSD in the resting state had already been suggested, before the elucidation of their crystal structures (Durell *et al.*, 1998, 2004; Tiwari-Woodruff *et al.*, 2000;

Gandhi & Isacoff, 2002, 2004). However, disclosure of several channel structures at atomic resolution, although only in the open/inactivated state, now allowed the involvement of computational approaches such as molecular dynamics (MD) simulations, and provided better estimates of the closed state and the voltage gating process at the molecular level.

6.1 - The resting state structure of Kv1.2 VSD from *ab initio* modeling and MD simulations

The first model of a voltage-gated channel in the resting state derived from structural data was obtained by Catterall's lab using *ab initio* Rosetta modeling on the Kv1.2 channel structure (Yarov-Yarovoy *et al.*, 2006). The model was based on structural restraints and established experimental data, such as the distance between the first gating charge R1 and the negative residues E226 on S2 and D259 on S3, or the exposure of the gating charges R3 and R4 to the intracellular water-accessible vestibule. The outward movement of the S4 segment that was produced by VSD activation from the closed state model to the activated state of the crystallographic structure was incredibly small, only $\sim 3 \text{ \AA}$ (and a clockwise rotation of $\sim 180^\circ$ on its axis) (Yarov-Yarovoy *et al.*, 2006). A comparable *ab initio* modeling procedure, in conjunction with MD simulations, was used by Isacoff's lab to define the resting state of Kv1.2 VSD (Pathak *et al.*, 2007). In addition to the standard experimental constraints, they took into account data from fluorescence scans on the local motion of external portions of the Shaker channel protein associated with specific activation steps. While the overall conclusions on the voltage sensor activation mechanism were similar to those proposed by Catterall's lab, Isacoff and coworkers' results differed in predicting a significantly larger S4 translocation ($6\text{--}8 \text{ \AA}$).

Starting from the closed state model of Kv1.2 developed in Isacoff's lab, Khalili-Araghi *et al.* made MD simulations in an explicit membrane and surrounding solvent, including polar lipid headgroups, to refine the resting state model (Khalili-Araghi *et al.*, 2010). In all simulations they found the formation of salt bridges between the gating charges on S4 and the highly conserved negative charges in S2 and S3 (interactions with polar lipids at the membrane-solution interface were also found). Their simulations also indicated a more inward position for S4 in the resting state than previously suggested by Isacoff and coworkers (Pathak *et al.*, 2007). Because of the focused transmembrane electric field fell within a distance not larger than $\sim 10 \text{ \AA}$, due to the water-filled vestibules on both sides of the VSD and the water-impermeant hydrophobic plug in between, they estimated a movement of S4 of $\sim 7 \text{ \AA}$ as sufficient to account for a charge of $12\text{--}14e_0$ transported across the membrane upon full VSD activation.

Khalili-Araghi *et al.* (2010) also found that a portion of S4 α -helix containing ~ 10 residues, located across the catalytic center, spontaneously transformed to a 3.10-helix while reaching the resting conformation. The 3.10-helix conformation would bring the gating arginine residues (R1–R4) on S4 to align vertically on one face of the helix, favoring salt bridge formation with acidic residues in S2 and S3 and stabilizing energetically the resting state. The presence of a 3.10-helix stretch in the lower portion of the S4 segment (~ 11 residues) was already observed in the open-state X-ray structures of Kv1.2 channels (Long *et al.*, 2007). Catterall's group also observed the S4 segments to adopt a 3.10-helix conformation in the bacterial Nav channel NaChBac, and suggested that charge pairing between the gating charges and the acidic residues forming the external and internal negative clusters that occur while S4 passes through the catalytic center would be facilitated by a transient shift of a stretch of S4 from α -helix into a 3.10-helix (DeCaen *et al.*, 2009).

A significant portion of the S4 segment was also found in the 3.10 helix conformation by Lindahl's group when they modeled with all-atom molecular dynamics the transition from the open X-ray structure of Kv1.2/2.1 chimera to the resting state, under the influence of a negative membrane potential (Bjelkmar *et al.*, 2009). Lindahl and coworkers further showed that the 3.10-helix transition that occurred between Q1 and R3 reduced the free energy associated with the initial steps of S4 movements towards the resting state (Schwaiger *et al.*, 2011).

Using MD simulations on Kv1.2 channel (Delemotte *et al.*, 2011) found a sequential translocation of the voltage sensor gating charges through the GCTC, characterized by its highly conserved phenylalanine F233, to reach the resting state of the channel that was characterized by all the gating charges having slipped below F233. Similar results were reported by Elinder and coworkers using the Cd²⁺ bridges strategy on Shaker channel to estimate the position of S4 and its interaction with the other segments of the VSD in several states, including the closed state(s), (Henrion *et al.*, 2012). They found that, at very hyperpolarizing conditions all the gating charges, including R1, becomes stabilized below F290. At a slightly milder hyperpolarization R1 is instead found above F290, interacting through salt bridge with E283.

These models of the resting state were found to strongly converge with the resulting picture that all the gating charges on S4 were pushed inwards below the hydrophobic plug, except arginine R1 that remained above F290, stably interacting with E1. We should recall that MacKinnon's lab had instead placed R1 below F290 in the resting state of Shaker, stabilized into the catalytic center (GCTC) by the acidic residues E293 (E2) in S2 and D316 (D3) in S3 (Tao *et al.*, 2010). Their view found experimental support from the Zn²⁺ bridging experiments where the double mutant I287H on S2 and R1H in Shaker channel would allow the formation of Zn²⁺ metal bridges in the resting state, suggesting that R1H had shifted further down, past F290, and bound to the GCTC (Lin *et al.*, 2011).

Figure 6.1 - The Consensus model of the resting state of Kv1.2 channels' VSD. A) VSD of Kv1.2 in the active state, shown for reference. The spheres represent the C α atoms of E1 (E226) and E2 (E236) on S2, and R1 (R294) on S4. **B)** Superposed resting-state VSD models from from (Pathak *et al.*, 2007; Delemotte *et al.*, 2011; Vargas *et al.*, 2011; Jensen *et al.*, 2012; Henrion *et al.*, 2012). Color code of the four helices: S1 gray, S2 yellow, S3 red, and S4 blue. [From Vargas *et al.*, 2012].

This discrepancy among different studies with regard to the position of R1 at rest may be due to the likelihood of there being more than one 'resting' state, each of which is occupied in relation to how much negative voltage is applied, and may not have been the same in the various studies. According to this view, the consensus model of (Vargas *et al.*, 2011) may represent the resting

state more populated at intermediate hyperpolarizations (the ‘penultimate resting state’, as it was termed by (Lin *et al.*, 2011)), whereas the resting state reported by (Tao *et al.*, 2010) and (Delemotte *et al.*, 2011), with R1 in the GCTC, can only be reached with strong hyperpolarizations. An additional consideration that could help reconcile these differing results is that most of the studies placing R1 above F290 were carried out upon mutating R1 into a neutral residue. It may be argued that upon neutralizing R1, S4 would hardly be able to reach its most inward position because it is just on the charged R1 where the transmembrane voltage acts to pull S4 fully inwards.

6.2 - The first resting structures of the Na channel’s VSD are provided by cryo-EM

In 2019, Catterall’s lab reported the high resolution cryo-EM and X-ray structure of the resting state of the bacterial Na channel NavAb, which they stabilized in this state by introducing mutations that were previously shown to shift the activation $V_{1/2}$ of Na channel to +60 mV (with the result that at 0 mV, the condition experienced during the cryo-EM procedure the channel – a tetramer of identical subunits – was in the resting state) (Wisedchaisri *et al.*, 2019)⁴. Moreover, they introduced disulfide crosslinks to lock the channel structure in the desired – resting (mutated channels) or activated (WT) – conformation (Lopez *et al.*, 1991; Gamal El-Din *et al.*, 2013). The stabilization of these states in these mutants were verified in functional studies to ensure that the structural data have clear functional correlates.

Analysis of these constructs provided critical insights into the structural rearrangements associated with the gating transitions of the channel’s VSD from the resting to the activated state, and back again. These included an inward displacement of ~ 11.5 Å for the S4 segment, and a significant rotation, with the gating charges interacting with different ion pair partners. In the resting state the first gating charge, R1, was found above the gating pore. With the resting and activated states of NavAb available, Catterall and coworkers could also model the putative transitions during channel activation. It was suggested that the outward movement of S4 segment occurring during activation translocates three gating charges, R2, R3 and R4, through the hydrophobic plug/transmembrane electric field. These conformation transitions were compatible with the classical Sliding helix model in which the S4 segment moves, while rotating along its axis. The gating charges, not directly exposed to the lipid hydrocarbon, interacted by forming sequential salt bridges with the acidic intracellular carboxyl-terminal domains as originally proposed by Clay Armstrong (Armstrong, 1981). Moreover, the transmembrane potential driving the gating charge translocation through the gating pore is concentrated over a narrow region of ~ 10 Å, which includes the evolutionary conserved phenylalanine residue (Wisedchaisri *et al.*, 2019).

⁴ Early 2019 Payandeh’s lab had reported two cryo-EM structures of intermediate, ‘non-activated’ states of the voltage sensor of Nav1.7 channels they attained using neurotoxins that bind to the voltage sensor of the fourth channel domain, and prevent its full activation until well above 0 mV (the voltage channels are subject during the procedure for EM) (Xu *et al.*, 2019; Clairfeuille *et al.*, 2019). Comparing their structures with the activated structure, without the toxin, they calculated ~ 10 Å and ~ 13 Å translation of the S4 helix. Both studies indeed focused on investigating the pharmacology of the α -scorpion toxin AaH2 on Na channels fast inactivation, and the structural glimpse on these non-activated states of the channel are more properly a byproduct of their investigation.

6.3 - The *Five-state gating* model

Using a combination of MD simulations and biased MD to visualize the conformation changes of the Kv1.2 VSD during channel deactivation (i.e., under hyperpolarized potentials), Delemotte *et al.* (2011) found five available states the channel could occupy during the deactivation process (Box 5). In addition to the open (starting) state (α) and the resting (final) state (ϵ) the channels would dwell in three intermediate states (β , γ and δ). While moving, the S4 segment would sequentially establish and break ion pairs with nearby negatively charged residues and lipid head groups, in a zipper-like fashion, as previously suggested. The positions of the gating charges on S4 in the resting (ϵ) state were congruent with those found by Pathak *et al.* (2007) using *ab initio* modeling, and by Khalili-Araghi *et al.* (2010) obtained with MD simulations. They also assessed the gating charge translocated during the entire transition process, from the open to the closed state, and found it to amount to $\sim 12.0e_0$, in good agreement with major biophysical studies ($12\text{--}14e_0$) (Schoppa *et al.*, 1992; Aggarwal & MacKinnon, 1996; Seoh *et al.*, 1996). The simulations also showed that in one of the four VSD the passage of the gating charges through the catalytic center involved the switching of a short stretch encompassing the charged residue that is crossing the gating pore into a 3.10-helix conformation, in a manner similar to the proposition advanced by Catterall's group (DeCaen *et al.*, 2009). As for the other VSDs, the passage of charges through the catalytic center occurred without any structural change of the α -helix conformation of the S4 segment (Delemotte *et al.*, 2011).

Elinder's lab made a significant contribution to the ongoing debate by presenting a whole voltage-sensor gating cycle comprised of one open and four closed states (Henrion *et al.*, 2012) (Box 5). They studied 20 specific interactions between S4 and the other segments of the VSD, in the framework of the five states that resulted from metal (Cd^{2+}) ion-bridge studies with cysteines. Models for each state were generated by Rosetta modeling (with no assumptions on the spatial position of the VSD helices from other studies), and finally refined by repeated MD simulations (Henrion *et al.*, 2012). According to this study, the S4 segment moves by no less than 12 Å from the active state (O) to the C3 state (the equivalent of the δ state of Delemotte *et al.* (2011), in this motion shifting three charges across the full membrane voltage drop. A deeper resting (closed) state C4 of the VSD (the ϵ state of Delemotte *et al.* (2011)) could be reached with very large hyperpolarizations (in which case, the S4 would move by ~ 17 Å). As already reported, Henrion *et al.*, (2012) also found S4 α -helix transiently switching into a 3.10-helix, over a segment of ~ 10 amino acids that maintains its location across the hydrophobic region, i.e. around F290, of the VSD. This helix conformation switch would remove the need of a physical rotation of the entire S4 α -helix, and minimize the energetics of the transitions (see below).

BOX 5 – The *Five-state gating* model

K channel deactivation has been shown to encompass five stable states of the VSD, with the voltage sensor moving inward by 10–15 Å, and the gating charges on S4 sequentially engaging in forming and breaking ion pairs with acidic residues of the VSD external vestibule and with the GCTC, comprising the phenylalanine F233 and the two acidic residues on S2 and S3.

Figure B5A - The Five-state gating model of Delemotte *et al.*, 2011. Representative conformations (α , β , γ , δ , and ϵ) of the VSDs resulting from MD simulations showing the interactions of the S4 basic residues (blue sticks: R1, R2, R3, R4, K5, and R6) with their privileged binding sites (red sticks: E183, E226, D259, E236), as well as with the lipid head group (yellow spheres, PO₄⁻). The highly conserved residue F233 of S2 is shown as cyan spheres. [From Delemotte *et al.*, 2011)].

Figure B5B - The Five-state gating model of Henrion *et al.*, 2012). Overlay of simulated VSD structures of the Kv1.2 channel after imposing harmonic restraints. Cartoon of the molecular models of VSD states during the gating process. [From Henrion *et al.*, 2012)]

In the years that followed the elucidation of the crystal structures of Kv1.2 and Kv1.2/2.1 chimera, both depicting the VSD in the activated state, there commenced intense research activity using multiple approaches aimed at understanding the position of the voltage sensor in the resting state and its interactions with the surroundings (Campos *et al.*, 2007; Pathak *et al.*, 2007; DeCaen *et al.*, 2008, 2009, 2011; Yarov-Yarovoy *et al.*, 2012; Henrion *et al.*, 2012). Notable in this context are the long MD simulations made in Shaw's lab to visualize a complete translocation of a K channel voltage sensor from the active state to the resting state, upon membrane hyperpolarizations (Jensen *et al.*, 2012).

7. Other approaches to modelling channel gating

MD simulation represents a powerful approach to investigate at atomic resolution the conformational transitions occurring in channel gating. It is however still computationally very expensive, and simulation timescales present a major challenge. MD simulation also falls short on the grounds of model validation, that is, the assessment of the extent to which its output reproduces the experimental results. To overcome these limitations alternative strategies have emerged over the years that simplify the system studied by applying sensible approximations, without impacting on the prospect of getting basic information on the dynamics of the gating structures. Importantly, these approaches also provide models that can be validated.

7.1 - Alternative approaches to investigate voltage sensor gating

We recall here the mesoscale model of Peyser and Nonner where the voltage sensor was represented by point charges immersed in a homogeneous dielectric environment, and statistical mechanics were applied to determine the stability of the various gating states (Peyser & Nonner, 2012a, 2012b). Model behavior appeared very robust and highly predictive of experimental data. Unfortunately, the model would only operate at equilibrium, and thus was unable to predict the dynamic features of macroscopic gating currents.

Another macroscopic approach to describe voltage gating was proposed by Wharshel's group (Dryga et al., 2012a, 2012b; Kim & Warshel, 2014). They used meta dynamics-based algorithms to find a reasonable energy profile for the movement of the voltage sensor during activation. With the energy profile known they predicted the dynamics of the voltage sensor using the Langevin equation, and the gating currents from the voltage sensor movement and the voltage profile across the VSD. The model was able to predict several essential features of the gating current observed experimentally, such as the fast gating component and the rising phase present at the beginning of the depolarizing pulse at relatively high membrane potentials.

Horng et al. recently proposed another model of voltage gating that assesses self-consistently, using a Poisson-Nernst-Planck formalism, the electrostatic energy resulting from the combination of the gating charges and the applied voltage (Horng et al., 2019). The gating charges, modeled as charged particles connected to the S4 segment by springs, would move by electrodiffusion and provide the force for pulling S4 across the VSD. This model is also capable of reproducing the main features of the experimental gating current. Shortcomings of the model are that it does not include the fixed charges present in the VSD (that represent the counter-charges for the gating charges on the S4 segment and regulate its movement), and the gating charge distribution and the geometry of the voltage sensor were not derived from available crystal structures.

Our lab has also developed a model of voltage gating, based on both the Poisson-Nernst-Planck formalism for the description of ion electrodiffusion and the assessment of the electrostatic potential, and the Brownian dynamics for the description of the voltage sensor movements (Catacuzzeno & Franciolini, 2019; Catacuzzeno et al., 2020). The geometry and charge distribution of the model were derived from a Shaker atomic structure obtained by homology modelling from the crystallographic structure of Kv1.2/Kv2.1 chimera (Long et al., 2007). The model fully agrees with recent MD and structural data indicating that during activation the voltage sensor visits five states (cf. Delemotte et al., 2011; Henrion et al., 2012) and translocates during its motion the first four gating charges across the gating pore (Figure 7.1A).

As result, the energy profile seen by the voltage sensor during activation, assessed self-consistently with the Poisson equation, displays four barriers (for the four gating charges on the voltage sensor relevant in the gating process) to cross (Figure 7.1C). The simulated gating currents obtained with our Brownian model reproduced all the main features of the gating currents recorded from typical K channels, and importantly provided a physical explanation for them (Catacuzzeno *et al.*, 2021b, 2021a)(Figure 7.1B).

Figure 7.1 – A) Schematics showing the geometry of the VSD of our model and the charges present in the S4 segment and in the rest of the VSD **B)** A family of simulated gating current obtained in response to depolarizing pulses. **C)** Energy profile experienced by the voltage sensor during its movement, at 0 mV. The energy profile is self-consistently assessed with the Poisson equation, considering the effect of all the charges present in the system. [From Catacuzzeno et al., 2021].

Our Brownian model was also able to accurately reproduce the gating current fluctuations (noise) from which individual charge packages transported by the voltage sensor during activation can be assessed. This is critical for readdressing and reinterpreting the classical experimental data on gating current noise obtained before the channel structures were known, which suggested shot current charge packages of $1.0e_0$. This has helped resolve the evident conflict by suggesting that the relatively high charge shot deduced from the fluctuation analysis actually results from a limited recording bandwidth (sequential gating charges could cross the gating pore in such rapid succession to become individually indistinguishable, and appear as a single larger charge (multiple charge crossing)) (Catacuzzeno *et al.*, 2021a).

8. Conclusions and outlook

In their landmark papers describing the ionic basis of the action potential, Hodgkin and Huxley predicted the gating charged particles that needed to move across the membrane to account for the voltage-dependent Na and K permeability changes found in the squid giant axon. The predicted gating currents were observed some 20 years later (Armstrong & Bezanilla, 1973; Schneider & Chandler, 1973; Keynes & Rojas, 1974), and estimated to result from the outward movement of $12\text{--}14e_0$ across the electric field to gate a single Na channel. Gating currents from K channels were later recorded and found to share essentially the same properties of Na gating currents. When the primary sequences of voltage-gated Na and K channels were elucidated, in early 1980s, and their architecture deduced, the fourth transmembrane domain, the S4 segment, was systematically found to contain an unexpected high number of positively charged residues, arginine mainly, and thus proposed to be the channels' voltage sensor (the structural counterpart of the gating charged particles of Hodgkin and Huxley). The first model of channel gating appeared in the form of the *Sliding helix* that postulated the outward, rotational movement of the putative voltage sensor, the S4 segment, with the gating charges sequentially forming ion pairs with the surrounding, to make the process energetically viable (Guy & Seetharamulu, 1986; Catterall, 1986). The first reliable high-resolution structures of crystallized voltage-gated ion channels (the Kv1.2, soon followed by the Kv1.2/Kv2.1 chimera; (Long *et al.*, 2005a, 2005b, 2007)), were both in the open state, with the S4 segment projected outwards, as expected from transmembrane

voltage zeroing of crystallization procedure. These atomic-resolution structures of voltage-gated channels allowed the involvement of MD modeling and thus greatly contribute to our understanding of voltage dependent gating, and to picture the the channel's resting state.

Brownian modeling, which simplified the systems studied by introducing sensible approximations as a tradeoff for creating the much needed macroscopic gating currents and their relevant parameters.

Outlook. Turning to future challenges in the field, the first goal is to obtain high resolution cryo-EM or X-ray crystallographic structures of each relevant functional and structural state of the channel. MD and other modeling approaches have so far tried to compensate for this limitation by picturing the unseen (intermediate) conformational states. Now we need to have more direct evidence and knowledge of them, which is an absolute requirement to visualize the actual rearrangements of VSD between the various gating states at the atomic level. This would also permit resolution of the laws of physics that guide these rearrangements, and attempt to provide a mechanistic interpretation of the full gating process. The main challenge in this context is to develop strategies to trap the VSD in the various gating states, and structural techniques to resolve for each of them the atoms' position at sub-angstrom level.

A second point is to reinstate the importance of energetics in channel gating, which is presently much undervalued. The complex trajectories of the gating charge translocations among the various gating states follow predefined paths determined by the energy landscapes encountered. Knowledge of these pathways and charting the energy profile would greatly help our understanding of the physical and molecular determinants of gating. Unfortunately they are poorly known at present, as they are shaped by the structural conformations of the VSD in its various states. Future studies should start trying to interpret data through VSD models that strictly correlate function and structure. Some of these models have begun to appear, that assess the electrostatic energy landscape seen by the voltage sensor using the Poisson's equation and considering all the charges present in the 3D structure of the Shaker VSD (Catacuzzeno & Franciolini, 2019).

Thirdly, we would encourage expansion of the use of multi-scale approaches in the study of channel gating. We are aware of the limits of MD modeling with regard to simulations duration and output verification. Brownian models overcome these limits, yet they are not immune from others, for instance determining critical parameters needed in their own modelling. To unite the forces of the two models, in a recent study we used a multi-scale hierarchical approach to test if Nav and Kv channels respond with different activation rate to potential change mainly because of the different polarizability in the gating pore (determined by whether there is a threonine or an isoleucine in the channel at equivalent position of 287 of Shaker). We first performed all-atom MD simulations on the atomic structure of Shaker VSD to assess water accessibility and effective dielectric constant in the VSD of both the wild-type and the silico mutated I287 with threonine (I287T), as in Nav channel. The results of these MD computations were used to assess the polarizability of the environment surrounding the voltage sensor needed for applying our Brownian model. This multi-scale approach allowed to reproduce the effects of the isoleucine to threonine mutation of residue 287 on the time course of the gating currents and in turn on the

firing threshold. Details can be found in the original papers of (Catacuzzeno & Franciolini, 2019; Catacuzzeno *et al.*, 2021b).

Finally, we think more consideration should be given to electro-diffusive theories in modeling channel gating. Most experimental data on voltage dependent gating accumulated over the past 50 years have been interpreted within the framework of rate models, an approach pioneered by Hodgkin and Huxley in their quantitative model of the gating charge movement during channel activation. Although this approach has been undoubtedly very useful to understand many key points about voltage dependent gating, it is now clear that its application to channel gating might not be always appropriate, or be the best choice. The structural and functional information accumulated over the years show that voltage dependent gating, namely the movement of the S4 segment across the membrane electric field, has many features of an electro-diffusive process. It can be approximated by a rate model only if the diffusive particle experience an energy profile characterized by a relatively high (i.e. over 4-5 kT) energy barrier separating the stable states. Our Brownian model of voltage gating, as well as information from MD simulations inform us that this condition may not always be present in real channels. In addition, rate models require only the rate constants as parameters connecting the different states, and thus they are not able to tell us much about the structural basis of the voltage dependent gating. Thus electro-diffusive models where the 3D structure of the VSD is more explicitly considered will certainly be more useful.

Acknowledgements: We are grateful to Bob Eisenberg and Pancho Bezanilla for the several discussions we had on the subject, and to Sandy Harper for reading and commenting on drafts of the Ms. Our original research has been supported over the years by grants from Cassa di Risparmio di Perugia.

References

- Aggarwal SK & MacKinnon R (1996). Contribution of the S4 segment to gating charge in the Shaker K⁺ channel. *Neuron* **16**, 1169–1177.
- Almers W (1978). Gating currents and charge movements in excitable membranes. *Rev Physiol Biochem Pharmacol* **82**, 96–190.
- Almers W & Armstrong CM (1980). Survival of K⁺ permeability and gating currents in squid axons perfused with K⁺-free media. *J Gen Physiol* **75**, 61–78.
- Armstrong CM (1971). Interaction of tetraethylammonium ion derivatives with the potassium channels of giant axons. *J Gen Physiol* **58**, 413–437.
- Armstrong CM (1981). Sodium channels and gating currents. *Physiol Rev* **61**, 644–683.
- Armstrong CM & Bezanilla F (1973). Currents related to movement of the gating particles of the sodium channels. *Nature* **242**, 459–461.
- Armstrong CM & Bezanilla F (1977). Inactivation of the sodium channel. II. Gating current experiments. *J Gen Physiol* **70**, 567–590.
- Armstrong CM & Wm Gilly F (1979). Fast and slow steps in the activation of sodium channels. *J Gen Physiol* **74**, 691–711.
- Bezanilla F (2002). Voltage sensor movements. *J Gen Physiol* **120**, 465–473.
- Bezanilla F (2008). How membrane proteins sense voltage. *Nat Rev Mol Cell Biol* **9**, 323–332.
- Bezanilla F, Perozo E & Stefani E (1994). Gating of Shaker K⁺ channels: II. The components of gating currents and

- a model of channel activation. *Biophys J* **66**, 1011–1021.
- Bezanilla F & Stefani E (1994). Voltage-dependent gating of ionic channels. *Annu Rev Biophys Biomol Struct* **23**, 819–846.
- Bjellmar P, Niemelä PS, Vattulainen I & Lindahl E (2009). Conformational changes and slow dynamics through microsecond polarized atomistic molecular simulation of an integral Kv1.2 ion channel. *PLoS Comput Biol*; DOI: 10.1371/JOURNAL.PCBI.1000289.
- Campos F V., Chanda B, Beirão PSL & Bezanilla F (2007). beta-Scorpion toxin modifies gating transitions in all four voltage sensors of the sodium channel. *J Gen Physiol* **130**, 257–268.
- Catacuzzeno L & Franciolini F (2019). Simulation of Gating Currents of the Shaker K Channel Using a Brownian Model of the Voltage Sensor. *Biophys J* **117**, 2005–2019.
- Catacuzzeno L, Franciolini F, Bezanilla F & Eisenberg RS (2021a). Gating current noise produced by Brownian models of a voltage sensor. *Biophys J* **120**, 3983–4001.
- Catacuzzeno L, Sforna L & Franciolini F (2020). Voltage-dependent gating in K channels: experimental results and quantitative models. *Pflugers Arch* **472**, 27–47.
- Catacuzzeno L, Sforna L, Franciolini F & Eisenberg RS (2021b). Multiscale modeling shows that dielectric differences make NaV channels faster than KV channels. *J Gen Physiol*; DOI: 10.1085/JGP.202012706.
- Catterall WA (1986). Molecular properties of voltage-sensitive sodium channels. *Annu Rev Biochem* **55**, 953–985.
- Cha A & Bezanilla F (1998). Structural implications of fluorescence quenching in the Shaker K⁺ channel. *J Gen Physiol* **112**, 391–408.
- Cha A, Snyder GE, Selvin PR & Bezanilla F (1999). Atomic scale movement of the voltage-sensing region in a potassium channel measured via spectroscopy. *Nature* **402**, 809–813.
- Chanda B, Asamoah OK, Blunck R, Roux B & Bezanilla F (2005). Gating charge displacement in voltage-gated ion channels involves limited transmembrane movement. *Nature* **436**, 852–856.
- Chanda B & Bezanilla F (2002). Tracking voltage-dependent conformational changes in skeletal muscle sodium channel during activation. *J Gen Physiol* **120**, 629–645.
- Clairfeuille T, Cloake A, Infield DT, Llongueras JP, Arthur CP, Li ZR, Jian Y, Martin-Eauclaire MF, Bougis PE, Ciferri C, Ahern CA, Bosmans F, Hackos DH, Rohou A & Payandeh J (2019). Structural basis of α -scorpion toxin action on Na^v channels. *Science*; DOI: 10.1126/SCIENCE.AAV8573.
- Colenso CK, Cao Y, Sessions RB, Hancox JC & Dempsey CE (2014). Voltage sensor gating charge transfer in a hERG potassium channel model. *Biophys J* **107**, L25–L28.
- Crouzy SC & Sigworth FJ (1993). Fluctuations in ion channel gating currents. Analysis of nonstationary shot noise. *Biophys J* **64**, 68–76.
- DeCaen PG, Yarov-Yarovoy V, Scheuer T & Catterall WA (2011). Gating charge interactions with the S1 segment during activation of a Na⁺ channel voltage sensor. *Proc Natl Acad Sci U S A* **108**, 18825–18830.
- DeCaen PG, Yarov-Yarovoy V, Sharp EM, Scheuer T & Catterall WA (2009). Sequential formation of ion pairs during activation of a sodium channel voltage sensor. *Proc Natl Acad Sci U S A* **106**, 22498–22503.
- DeCaen PG, Yarov-Yarovoy V, Zhao Y, Scheuer T & Catterall WA (2008). Disulfide locking a sodium channel voltage sensor reveals ion pair formation during activation. *Proc Natl Acad Sci U S A* **105**, 15142–15147.
- Delemotte L, Tarek M, Klein ML, Amaral C & Treptow W (2011). Intermediate states of the Kv1.2 voltage sensor from atomistic molecular dynamics simulations. *Proc Natl Acad Sci U S A* **108**, 6109–6114.
- Dryga A, Chakrabarty S, Vicatos S & Warshel A (2012a). Coarse grained model for exploring voltage dependent ion channels. *Biochim Biophys Acta* **1818**, 303–317.
- Dryga A, Chakrabarty S, Vicatos S & Warshel A (2012b). Realistic simulation of the activation of voltage-gated ion channels. *Proc Natl Acad Sci U S A* **109**, 3335–3340.
- Durell SR, Hao Y & Guy HR (1998). Structural models of the transmembrane region of voltage-gated and other K⁺ channels in open, closed, and inactivated conformations. *J Struct Biol* **121**, 263–284.
- Durell SR, Shrivastava IH & Guy HR (2004). Models of the structure and voltage-gating mechanism of the shaker K⁺ channel. *Biophys J* **87**, 2116–2130.
- Gamal El-Din TM, Martinez GQ, Payandeh J, Scheuer T & Catterall WA (2013). A gating charge interaction required for late slow inactivation of the bacterial sodium channel NavAb. *J Gen Physiol* **142**, 181–190.
- Gandhi CS & Isacoff EY (2002). Molecular models of voltage sensing. *J Gen Physiol* **120**, 455–463.
- Glauner KS, Mannuzzu LM, Gandhi CS & Isacoff EY (1999). Spectroscopic mapping of voltage sensor movement in the Shaker potassium channel. *Nature* **402**, 813–817.
- Gosselin-Badaroudine P, Delemotte L, Moreau A, Klein ML & Chahine M (2012). Gating pore currents and the resting state of Nav1.4 voltage sensor domains. *Proc Natl Acad Sci U S A* **109**, 19250–19255.

- Greenblatt RE, Blatt Y & Montai M (1985). The structure of the voltage-sensitive sodium channel. Inferences derived from computer-aided analysis of the *Electrophorus electricus* channel primary structure. *FEBS Lett* **193**, 125–134.
- Guy HR & Seetharamulu P (1986). Molecular model of the action potential sodium channel. *Proc Natl Acad Sci U S A* **83**, 508–512.
- Henrion U, Renhorn J, Börjesson SI, Nelson EM, Schwaiger CS, Bjelkmar P, Wallner B, Lindahl E & Elinder F (2012). Tracking a complete voltage-sensor cycle with metal-ion bridges. *Proc Natl Acad Sci U S A* **109**, 8552–8557.
- Hille B (1968). Pharmacological modifications of the sodium channels of frog nerve. *J Gen Physiol* **51**, 199–219.
- Hirschberg B, Rovner A, Lieberman M & Patlak J (1995). Transfer of twelve charges is needed to open skeletal muscle Na⁺ channels. *J Gen Physiol* **106**, 1053–1068.
- Hodgkin AL & Huxley AF (1952a). The dual effect of membrane potential on sodium conductance in the giant axon of *Loligo*. *J Physiol* **116**, 497.
- Hodgkin AL & Huxley AF (1952b). The components of membrane conductance in the giant axon of *Loligo*. *J Physiol* **116**, 473.
- Hodgkin AL & Huxley AF (1952c). Currents carried by sodium and potassium ions through the membrane of the giant axon of *Loligo*. *J Physiol* **116**, 449.
- Hodgkin AL & Huxley AF (1952d). A quantitative description of membrane current and its application to conduction and excitation in nerve. *J Physiol* **117**, 500.
- Horng TL, Eisenberg RS, Liu C & Bezanilla F (2019). Continuum Gating Current Models Computed with Consistent Interactions. *Biophys J* **116**, 270–282.
- Islas LD & Sigworth FJ (1999). Voltage sensitivity and gating charge in Shaker and Shab family potassium channels. *J Gen Physiol* **114**, 723–741.
- Jensen M, Jogini V, Borhani DW, Leffler AE, Dror RO & Shaw DE (2012). Mechanism of voltage gating in potassium channels. *Science* **336**, 229–233.
- Jiang Y, Lee A, Chen J, Ruta V, Cadene M, Chait BT & MacKinnon R (2003a). X-ray structure of a voltage-dependent K⁺ channel. *Nature* **423**, 33–41.
- Jiang Y, Ruta V, Chen J, Lee A & MacKinnon R (2003b). The principle of gating charge movement in a voltage-dependent K⁺ channel. *Nature* **423**, 42–48.
- Kamb A, Tseng-Crank J & Tanouye MA (1988). Multiple products of the *Drosophila* Shaker gene may contribute to potassium channel diversity. *Neuron* **1**, 421–430.
- Katz B & Miledi R (1966). Input-output relation of a single synapse. *Nature* **212**, 1242–1245.
- Keynes RD & Rojas E (1974). Kinetics and steady-state properties of the charged system controlling sodium conductance in the squid giant axon. *J Physiol* **239**, 393–434.
- Khalili-Araghi F, Jogini V, Yarov-Yarovoy V, Tajkhorshid E, Roux B & Schulten K (2010). Calculation of the gating charge for the Kv1.2 voltage-activated potassium channel. *Biophys J* **98**, 2189–2198.
- Kim I & Warshel A (2014). Coarse-grained simulations of the gating current in the voltage-activated Kv1.2 channel. *Proc Natl Acad Sci U S A* **111**, 2128–2133.
- Koch WJ, Ellinor PT & Schwartz A (1990). cDNA cloning of a dihydropyridine-sensitive calcium channel from rat aorta. Evidence for the existence of alternatively spliced forms. *J Biol Chem* **265**, 17786–17791.
- Lacroix JJ & Bezanilla F (2011). Control of a final gating charge transition by a hydrophobic residue in the S2 segment of a K⁺ channel voltage sensor. *Proc Natl Acad Sci U S A* **108**, 6444–6449.
- Lee SY, Lee A, Chen J & MacKinnon R (2005). Structure of the KvAP voltage-dependent K⁺ channel and its dependence on the lipid membrane. *Proc Natl Acad Sci U S A* **102**, 15441–15446.
- Lin M chin A, Hsieh JY, Mock AF & Papazian DM (2011). R1 in the Shaker S4 occupies the gating charge transfer center in the resting state. *J Gen Physiol* **138**, 155–163.
- Logothetis DE, Kammen BF, Lindpaintner K, Bisbas D & Nadal-Ginard B (1993). Gating charge differences between two voltage-gated K⁺ channels are due to the specific charge content of their respective S4 regions. *Neuron* **10**, 1121–1129.
- Long SB, Campbell EB & MacKinnon R (2005a). Voltage sensor of Kv1.2: structural basis of electromechanical coupling. *Science* **309**, 903–908.
- Long SB, Campbell EB & MacKinnon R (2005b). Crystal structure of a mammalian voltage-dependent Shaker family K⁺ channel. *Science* **309**, 897–903.
- Long SB, Tao X, Campbell EB & MacKinnon R (2007). Atomic structure of a voltage-dependent K⁺ channel in a lipid membrane-like environment. *Nature* **450**, 376–382.

- Lopez GA, Jan YN & Jan LY (1991). Hydrophobic substitution mutations in the S4 sequence alter voltage-dependent gating in Shaker K⁺ channels. *Neuron* **7**, 327–336.
- MacKinnon R (1991). Determination of the subunit stoichiometry of a voltage-activated potassium channel. *Nature* **350**, 232–235.
- McCormack K, Joiner WI & Heinemann SH (1994). A characterization of the activating structural rearrangements in voltage-dependent Shaker K⁺ channels. *Neuron* **12**, 301–315.
- Meves H & Chandler WK (1965). Ionic Selectivity in Perfused Giant Axons. *J Gen Physiol* **48**, 31.
- Mikami A, Imoto K, Tanabe T, Niidome T, Mori Y, Takeshima H, Narumiya S & Numa S (1989). Primary structure and functional expression of the cardiac dihydropyridine-sensitive calcium channel. *Nature* **340**, 230–233.
- Moore EN (1967). Microelectrode studies on retrograde concealment of multiple premature ventricular responses. *Circ Res* **20**, 88–98.
- Noceti F, Baldelli P, Wei X, Qin N, Toro L, Birnbaumer L & Stefani E (1996). Effective gating charges per channel in voltage-dependent K⁺ and Ca²⁺ channels. *J Gen Physiol* **108**, 143–155.
- Noda M, Shimizu S, Tanabe T, Takai T, Kayano T, Ikeda T, Takahashi H, Nakayama H, Kanaoka Y, Minamino N, Kangawa K, Matsuo H, Raftery MA, Hirose T, Inayama S, Hayashida H, Miyata T & Numa S (1984). Primary structure of *Electrophorus electricus* sodium channel deduced from cDNA sequence. *Nature* **312**, 121–127.
- Papazian DM, Schwarz TL, Tempel BL, Jan YN & Jan LY (1987). Cloning of genomic and complementary DNA from Shaker, a putative potassium channel gene from *Drosophila*. *Science* **237**, 749–753.
- Pathak MM, Yarov-Yarovoy V, Agarwal G, Roux B, Barth P, Kohout S, Tombola F & Isacoff EY (2007). Closing in on the resting state of the Shaker K(+) channel. *Neuron* **56**, 124–140.
- Payandeh J, Scheuer T, Zheng N & Catterall WA (2011). The crystal structure of a voltage-gated sodium channel. *Nature* **475**, 353–359.
- Perozo E, MacKinnon R, Bezanilla F & Stefani E (1993). Gating currents from a nonconducting mutant reveal open-closed conformations in Shaker K⁺ channels. *Neuron* **11**, 353–358.
- Peyser A & Nonner W (2012a). Voltage sensing in ion channels: mesoscale simulations of biological devices. *Phys Rev E Stat Nonlin Soft Matter Phys*; DOI: 10.1103/PHYSREVE.86.011910.
- Peyser A & Nonner W (2012b). The sliding-helix voltage sensor: mesoscale views of a robust structure-function relationship. *Eur Biophys J* **41**, 705–721.
- Pless SA, Galpin JD, Niciforovic AP & Ahern CA (2011). Contributions of counter-charge in a potassium channel voltage-sensor domain. *Nat Chem Biol* **7**, 617–623.
- Pongs O, Kecskemethy N, Müller R, Krahe-Jentgens I, Baumann A, Kiltz HH, Canal I, Llamazares S & Ferrus A (1988). Shaker encodes a family of putative potassium channel proteins in the nervous system of *Drosophila*. *EMBO J* **7**, 1087–1096.
- Posson DJ, Ge P, Miller C, Bezanilla F & Selvin PR (2005). Small vertical movement of a K⁺ channel voltage sensor measured with luminescence energy transfer. *Nature* **436**, 848–851.
- Posson DJ & Selvin PR (2008). Extent of voltage sensor movement during gating of shaker K⁺ channels. *Neuron* **59**, 98–109.
- Ruta V, Jiang Y, Lee A, Chen J & MacKinnon R (2003). Functional analysis of an archaebacterial voltage-dependent K⁺ channel. *Nature* **422**, 180–185.
- Schneider MF & Chandler WK (1973). Voltage dependent charge movement of skeletal muscle: a possible step in excitation-contraction coupling. *Nature* **242**, 244–246.
- Schoppa NE, McCormack K, Tanouye MA & Sigworth FJ (1992). The size of gating charge in wild-type and mutant Shaker potassium channels. *Science* **255**, 1712–1715.
- Schwaiger CS, Bjelkmar P, Hess B & Lindahl E (2011). ₃10-helix conformation facilitates the transition of a voltage sensor S4 segment toward the down state. *Biophys J* **100**, 1446–1454.
- Seoh SA, Sigg D, Papazian DM & Bezanilla F (1996). Voltage-sensing residues in the S2 and S4 segments of the Shaker K⁺ channel. *Neuron* **16**, 1159–1167.
- Sigg D, Stefani E & Bezanilla F (1994). Gating current noise produced by elementary transitions in Shaker potassium channels. *Science* **264**, 578–582.
- Starace DM & Bezanilla F (2001). Histidine scanning mutagenesis of basic residues of the S4 segment of the shaker k⁺ channel. *J Gen Physiol* **117**, 469–490.
- Starace DM & Bezanilla F (2004). A proton pore in a potassium channel voltage sensor reveals a focused electric field. *Nature* **427**, 548–553.
- Stühmer W, Conti F, Suzuki H, Wang X, Noda M, Yahagi N, Kubo H & Numa S (1989). Structural parts involved in activation and inactivation of the sodium channel. *Nature* **339**, 597–603.

- Tanabe T, Takeshima H, Mikami A, Flockerzi V, Takahashi H, Kangawa K, Kojima M, Matsuo H, Hirose T & Numa S (1987). Primary structure of the receptor for calcium channel blockers from skeletal muscle. *Nature* **328**, 313–318.
- Tao X, Lee A, Limapichat W, Dougherty DA & MacKinnon R (2010). A gating charge transfer center in voltage sensors. *Science* **328**, 67–73.
- Taylor RE & Bezanilla F (1983). Sodium and gating current time shifts resulting from changes in initial conditions. *J Gen Physiol* **81**, 773–784.
- Tempel BL, Papazian DM, Schwarz TL, Jan YN & Jan LY (1987). Sequence of a probable potassium channel component encoded at Shaker locus of *Drosophila*. *Science* **237**, 770–775.
- Tiwari-Woodruff SK, Lin MCA, Schulteis CT & Papazian DM (2000). Voltage-dependent structural interactions in the Shaker K(+) channel. *J Gen Physiol* **115**, 123–138.
- Tombola F, Pathak MM & Isacoff EY (2005). Voltage-sensing arginines in a potassium channel permeate and occlude cation-selective pores. *Neuron* **45**, 379–388.
- Tytgat J & Hess P (1992). Evidence for cooperative interactions in potassium channel gating. *Nature* **359**, 420–423.
- Vargas E, Bezanilla F & Roux B (2011). In search of a consensus model of the resting state of a voltage-sensing domain. *Neuron* **72**, 713–720.
- White MM & Bezanilla F (1985). Activation of squid axon K⁺ channels. Ionic and gating current studies. *J Gen Physiol* **85**, 539–554.
- Wisedchaisri G, Tonggu L, McCord E, Gamal El-Din TM, Wang L, Zheng N & Catterall WA (2019). Resting-State Structure and Gating Mechanism of a Voltage-Gated Sodium Channel. *Cell* **178**, 993-1003.e12.
- Xu H, Li T, Rohou A, Arthur CP, Tzakoniati F, Wong E, Estevez A, Kugel C, Franke Y, Chen J, Ciferri C, Hackos DH, Koth CM & Payandeh J (2019). Structural Basis of Nav1.7 Inhibition by a Gating-Modifier Spider Toxin. *Cell* **176**, 702-715.e14.
- Yang N, George AL & Horn R (1996). Molecular basis of charge movement in voltage-gated sodium channels. *Neuron* **16**, 113–122.
- Yang N & Horn R (1995). Evidence for voltage-dependent S4 movement in sodium channels. *Neuron* **15**, 213–218.
- Yarov-Yarovoy V, Baker D & Catterall WA (2006). Voltage sensor conformations in the open and closed states in ROSETTA structural models of K(+) channels. *Proc Natl Acad Sci U S A* **103**, 7292–7297.
- Yarov-Yarovoy V, DeCaen PG, Westenbroek RE, Pan CY, Scheuer T, Baker D & Catterall WA (2012). Structural basis for gating charge movement in the voltage sensor of a sodium channel. *Proc Natl Acad Sci U S A*; DOI: 10.1073/PNAS.1118434109/-/DCSUPPLEMENTAL/SAPP.PDF.
- Zagotta WN, Hoshi T, Dittman J & Aldrich RW (1994). Shaker potassium channel gating. II: Transitions in the activation pathway. *J Gen Physiol* **103**, 279–319.